# Rank-Aware Spectral Bounds on Attention Logits for Stable Low-Precision Training

**Seyed Morteza Emadi** [1]

## Abstract

Attention scores in transformers are bilinear forms $S_{ij} = x_i^\top M x_j / \sqrt{d_h}$ whose maximum magnitude governs overflow risk in low-precision training. We derive a *rank-aware concentration inequality*: when the interaction matrix $M = W^Q W^{K\top}$ has rank $r \ll d$, tail probabilities for $\max_{i,j} |S_{ij}|$ decay as $\exp(-d^2\alpha^2/(\gamma r))$ rather than $\exp(-d\alpha^2)$, where $\gamma > 1$. For transformer attention where $r = d_h$, this yields $8$–$28\times$ tighter concentration than rank-agnostic bounds in modern architectures. We apply this result to FP8 training, deriving *geometry-aware scale factors* that provide principled overflow guarantees without observing activations. The method computes per-layer scales from the spectral norm $\|W^Q W^{K\top}\|_2$ via implicit power iteration, includes a grouped query attention formulation that avoids key expansion, and remains compatible with fused attention kernels. Across GPT-2 XL to Llama-2-70B, geometry-aware scaling eliminates overflows in transient scenarios where delayed scaling fails, while achieving comparable downstream MMLU accuracy.

## 1. Introduction

Attention logits in transformers exhibit a bilinear structure: each score $S_{ij} = x_i^\top W^Q W^{K\top} x_j / \sqrt{d_h}$ depends on the interaction of two projected vectors. This structure determines the maximum magnitude of attention logits, a quantity with direct consequences for low-precision training. FP8 formats reduce precision from 16 to 8 bits to enable substantial speedups on memory-bound workloads (Micikevicius et al., 2022), but the E4M3 format spans only

[1] UNC-Chapel Hill, Kenan-Flagler Business School, Chapel Hill, NC, USA. Correspondence to: Seyed Morteza Emadi <seyed_emadi@kenan-flagler.unc.edu>.

*Proceedings of the $43^{rd}$ International Conference on Machine Learning*, Seoul, South Korea. PMLR 306, 2026. Copyright 2026 by the author(s).

$\pm448$ compared to FP16's $\pm65{,}504$. Our focus is on the calibration problem that enables safe deployment: when attention logits exceed this range, quantization overflows produce NaN values that corrupt training.

**Problem formulation.** Let $R_{\max}$ denote the representable range of the target format ($R_{\max} = 448$ for E4M3). Given a scale factor $s > 0$, quantization overflows when $\max_{i,j} |S_{ij}| > s \cdot R_{\max}$. The calibration problem is to select $s$ that:

1. **(Safety)** Ensures $\Pr\big(\max_{i,j} |S_{ij}| > s \cdot R_{\max}\big) \leq \delta^*$ for a target failure probability $\delta^*$;

2. **(Utilization)** Maximizes precision utilization $\mathbb{E}[\max_{i,j} |S_{ij}|]/(s \cdot R_{\max})$, minimizing quantization error.

These objectives trade off: smaller $s$ improves utilization but increases overflow risk. The challenge is to calibrate $s$ *without observing* $\max_{i,j} |S_{ij}|$, which would require materializing the full attention matrix.

Existing methods derive scale factors from *observed* activation statistics, either from a history buffer (delayed scaling) or per-iteration (current scaling), without analyzing the bilinear mechanism that governs logit magnitudes.

**Delayed scaling and its limitations.** The standard approach for FP8 training is *delayed scaling* (Micikevicius et al., 2022), which maintains a history buffer of activation maxima (typically 16 steps) and derives scale factors from past observations:

$$\text{scale}_t = \frac{\max(\text{history})}{448 \cdot \eta}, \tag{1}$$

where $\eta < 1$ provides a safety margin. Scale factors are computed *before* the current forward pass, avoiding the need to observe current activations.

However, delayed scaling assumes activation magnitudes change slowly. This assumption breaks whenever weight dynamics outpace the history buffer. We call this failure mode *history staleness*. Common triggers include: (i) loading pretrained checkpoints, where FP8 history

initializes to defaults while weights reflect extensive prior training; (ii) resuming from saved states, since standard checkpointing omits scaling state; and (iii) learning rate transitions, where warmup or cyclic schedules cause weights to shift faster than history adapts. An alternative is *current scaling*, which computes scale factors from $\max |S_t|$ in each forward pass. This eliminates staleness but requires materializing the full $L \times L$ attention score matrix, incompatible with fused attention kernels like FlashAttention (Dao et al., 2022; Dao, 2024) that achieve $O(L)$ memory precisely by avoiding this materialization.

Table 1 summarizes this dilemma: delayed scaling is compatible with fused kernels but fails during transients; current scaling handles transients but requires materializing the score matrix. Prior to this work, no method achieved both.

*Table 1.* The FP8 scaling dilemma.

| Method | Transient-Safe | Fused-Compat. |
|---|---|---|
| Delayed | ✗ | ✓ |
| Current | ✓ | ✗ |
| **Ours** | ✓ | ✓ |

**Our approach: predictive calibration from weight geometry.** We propose a fundamentally different paradigm. Instead of *reacting* to observed activations (whether past or present), we *predict* their bounds from the singular value structure of the query-key interaction matrix. Attention scores form a bilinear structure:

$$S_{ij} = \frac{x_i^\top M x_j}{\sqrt{d_h}}, \quad \text{where } M = W^Q W^{K\top} \in \mathbb{R}^{d \times d} \quad (2)$$

is the query-key *interaction matrix* and $x_i, x_j$ are input embeddings. For *pre-LN architectures* (including GPT-2, Llama, and Mistral, which place normalization before attention), LayerNorm (or RMSNorm (Zhang & Sennrich, 2019)) constrains $\|x_i\| \approx \sqrt{d}$, yielding:

$$\max_{i,j} |S_{ij}| \leq \|M\|_2 \cdot \frac{d}{\sqrt{d_h}} =: B_{\max}. \quad (3)$$

This bound depends only on current weights, not activations, and can be estimated via power iteration (Golub & Van Loan, 2013) without forming the $d \times d$ matrix $M$. When weights change, the bound updates immediately, enabling *predictive* calibration that eliminates the failure modes of delayed scaling during transients such as checkpoint loading, resumption, and learning rate changes. Since we never observe activations, compatibility with fused attention kernels is preserved.

The worst-case bound $B_{\max}$ assumes inputs align perfectly with top singular vectors, which becomes exponentially unlikely as dimension increases. We therefore introduce a

calibration factor $\alpha \in (0, 1)$ to obtain a tighter bound $B_\alpha = \alpha \cdot B_{\max}$. To quantify the associated risk, we derive a *rank-aware concentration inequality* on $\Pr(\max_{i,j} |S_{ij}| \geq B_\alpha)$ that exploits the low-rank structure of the interaction matrix ($\text{rank}(M) = d_h \ll d$). The key insight is that although $M$ acts on $d$-dimensional space, its range is only $d_h$-dimensional; inputs are exponentially unlikely to find this low-dimensional subspace of maximum amplification. The resulting tail exponent improves from $d\alpha^2$ to $d^2\alpha^2/(\gamma d_h)$, a factor of $d/(\gamma d_h) = 8\text{--}28\times$ in modern architectures (where $\gamma > 1$ is set per architecture by the safety constraint of Section 3.2). This analysis provides a principled selection rule: given a target overflow probability $\delta^*$, we derive the minimum $\alpha$ guaranteeing $\Pr(\max_{i,j} |S_{ij}| \geq B_\alpha) \leq \delta^*$. This conservative $\alpha$ targets robustness in *transient-prone* workflows (initialization from a higher-precision checkpoint, resumption without preserved FP8 state, and early learning-rate warmup), in which a single overflow can corrupt or terminate training; for steady-state fine-tuning, an optional empirical calibration (Section 3.5) tightens $\alpha$ for utilization while retaining predictive scaling thereafter.

**Contributions.**

1. **Tight spectral interaction bound.** We prove $\max_{i,j} |S_{ij}| \leq \|W^Q W^{K\top}\|_2 \cdot d/\sqrt{d_h}$ under the distributional constraints induced by LayerNorm or RMSNorm. This bound is never looser than the naive submultiplicative bound $\|W^Q\|_2\|W^K\|_2$ and is strictly tighter unless the top right singular vectors align (Corollary 3.3). We show extension to RoPE architectures, with the tighter interaction bound validated empirically (Corollary 3.6).

2. **Rank-aware concentration inequality.** We prove a concentration inequality for bilinear forms $x_i^\top M x_j$ when $M$ has rank $r \ll d$ and inputs are approximately uniform on the sphere (Proposition 3.4). The key insight is a two-stage conditioning argument: first bound the probability that projections onto $M$'s row space are atypical, then apply concentration with a tighter Lipschitz constant. This yields tail exponents of order $d^2\alpha^2/(\gamma r)$ versus $d\alpha^2$ for rank-agnostic bounds, an improvement of $d/(\gamma r)$ that becomes $8\text{--}28\times$ in modern transformers where $r = d_h$. Applied to attention, this provides a principled selection rule: given a target overflow probability $\delta^*$, we derive the minimum calibration factor $\alpha$ guaranteeing safety.

3. **Implicit spectral norm estimator.** We estimate $\|W^Q W^{K\top}\|_2$ via power iteration without forming the $d \times d$ interaction matrix, achieving $O(n_{\text{heads}} \cdot d_h \cdot d)$ cost per layer. For grouped query attention, we derive an implicit formulation that avoids key matrix expansion (Section 4.2).

4. **Empirical validation.** Zero overflows across GPT-2 XL through Llama-2-70B on all transient scenarios where delayed scaling fails. Downstream evaluation on MMLU confirms preserved capability: our auto-$\alpha$ calibration achieves accuracy comparable to delayed scaling (32.7% vs. 32.6%) while eliminating all overflows, whereas conservative spectral bounds degrade to 28.7% due to under-utilization (Section 5.4).

**Positioning.** The spectral bound in Equation (3) builds on classical matrix analysis; our contribution lies in developing a complete calibration framework for low-precision training. This requires solving three problems that the bound alone does not address: (i) principled probabilistic calibration under the rank-deficient structure of attention, (ii) efficient per-layer computation via implicit power iteration, and (iii) an implicit GQA formulation that avoids key expansion. Together, these enable the first calibration method that is both robust to transients and compatible with fused attention kernels.

## 2. Related Work

**FP8 training and scaling strategies.** Micikevicius et al. (2022) introduced FP8 formats and delayed (history-based) scaling, now standard in low-precision training frameworks. DeepSeek-V3 (DeepSeek-AI, 2024) employs per-tile scaling (e.g., $1 \times 128$ blocks) but retains attention in BF16/FP32. Microscaling (Rouhani et al., 2023) similarly uses block-wise factors. These approaches address *spatial heterogeneity* within tensors but still derive scales from observed values, inheriting history staleness during transients. FlashAttention-3 (Shah et al., 2024) introduces native FP8 support with internal block quantization, but requires inputs cast to FP8 *before* kernel entry. Incorrect global scale factors cause overflow before internal mechanisms can compensate. Other recent work addresses orthogonal aspects: architectural modifications (Hernández-Cano et al., 2025), inference fallback (Lee et al., 2025), and LoRA fine-tuning (Choi et al., 2025). Our method provides a *safety envelope*: geometry-aware bounds ensure inputs are within safe range, and are orthogonal to block-wise optimizations.

**Predictive scaling.** Unit Scaling (Blake et al., 2023) eliminates scaling factors via careful initialization that maintains unit variance throughout the network, but requires training from scratch and cannot be applied to pretrained checkpoints. MOSS (Zhang et al., 2025) exploits the bounded update property of AdamW ($|\Delta W_t| \leq \eta$) to predict *weight* magnitudes without runtime max-reduction. However, MOSS requires the MXFP8 microscaling format with hardware-specific support. More fundamentally, bounds derived from individual weight norms $\|W^Q\|$ and $\|W^K\|$ are too loose for effective FP8 calibration: the naive bound $\|W^Q\|_2 \|W^K\|_2$ can substantially overestimate when

singular vectors of $W^Q$ and $W^K$ do not align, wasting dynamic range (Corollary 3.3). Our method addresses attention-specific overflow using only the standard E4M3 format, requires no custom kernels, and is compatible with NVIDIA's Transformer Engine.

**Spectral methods for transformer stability.** Zhai et al. (2023) show that *entropy collapse* (attention concentrating on single tokens) is governed by the spectral norm of the query-key interaction matrix. Our work reveals the same quantity governs numerical overflow risk, connecting optimization stability and numerical stability through shared geometric structure. Miyato et al. (2018) introduced spectral normalization for GANs as a regularizer; we use spectral norms for *prediction* rather than constraint.

## 3. Theoretical Foundation

This section develops the theoretical basis for geometry-aware scaling.

### 3.1. Spectral Bound on Attention Logits

Pre-softmax attention scores are computed as $S = QK^\top/\sqrt{d_h}$, where $Q = XW^Q$ and $K = XW^K$. Each score can be written as:

$$S_{ij} = \frac{x_i^\top M x_j}{\sqrt{d_h}}, \quad \text{where } M = W^Q W^{K\top} \in \mathbb{R}^{d \times d}. \quad (4)$$

The matrix $M$ is the *query-key interaction matrix*. Unlike FFN layers (additive structure) or embeddings (element-wise), attention's *bilinear* structure compounds magnitudes: large queries meeting large keys produce scores that can exceed the E4M3 threshold of 448. This makes attention the primary overflow bottleneck in FP8 training.

To bound attention logits, we exploit this bilinear structure. Proofs for this section are in Appendix A. A natural first approach tracks $\|W^Q\|_2$ and $\|W^K\|_2$ separately:

**Proposition 3.1** (Naive bound). *Let $\|x_i\|_2 \leq B_X$ for all tokens. Then:*

$$\max_{i,j} |S_{ij}| \leq \frac{\|W^Q\|_2 \|W^K\|_2 \cdot B_X^2}{\sqrt{d_h}}. \quad (5)$$

This bound is loose because it treats $W^Q$ and $W^K$ independently. Analyzing $M = W^Q W^{K\top}$ directly yields a tighter result:

**Proposition 3.2** (Interaction bound). *Under the same conditions:*

$$\max_{i,j} |S_{ij}| \leq \frac{\|W^Q W^{K\top}\|_2 \cdot B_X^2}{\sqrt{d_h}}. \quad (6)$$

**Corollary 3.3** (Interaction bound is tighter). $\|W^Q W^{K\top}\|_2 \leq \|W^Q\|_2 \|W^K\|_2$, *with equality iff the top right singular vectors of $W^Q$ and $W^K$ coincide.*

For pre-LN architectures (GPT-2, Llama, Mistral), LayerNorm or RMSNorm (Zhang & Sennrich, 2019) precedes attention, constraining $\|x_i\|_2^2 \approx d$ and giving $B_X = \sqrt{d}$. Substituting yields the *worst-case bound*:

$$\max_{i,j} |S_{ij}| \leq \|W^Q W^{K\top}\|_2 \cdot \frac{d}{\sqrt{d_h}} =: B_{\max}. \quad (7)$$

**Calibrating the bound.** The bound $B_{\max}$ assumes input vectors align perfectly with the top singular vectors of $M$. In practice, this is overly pessimistic: the worst case requires inputs to find the direction of maximum amplification in a $d$-dimensional space, but LayerNorm-normalized inputs behave like random directions on the sphere. In high dimensions, random vectors are nearly orthogonal to any fixed direction, making such alignment exponentially unlikely.

This concentration phenomenon motivates introducing a calibration factor $\alpha \in (0, 1)$. The *calibrated bound* is:

$$B_\alpha := \alpha \cdot B_{\max} = \alpha \cdot \|M\|_2 \cdot \frac{d}{\sqrt{d_h}}. \quad (8)$$

Smaller $\alpha$ yields a tighter bound and higher FP8 dynamic range utilization, but increases the probability that actual logits exceed the bound. The natural question is how to select $\alpha$: small enough to be practical, yet large enough to guarantee safety. In the next section, we develop a probabilistic framework that provides a principled answer.

### 3.2. Rank-Aware Probabilistic Guarantee

We derive a probabilistic bound on $\Pr(\max_{i,j} |S_{ij}| \geq B_\alpha)$ that enables principled selection of the calibration factor $\alpha$. Proofs for this section are in Appendix B.

**Assumption (Spherical token directions).** For Pre-LN transformers, we model post-LayerNorm (or post-RMSNorm) token vectors as $x_i = \sqrt{d} u_i$, where directions $u_i$ are approximately isotropic (idealized as i.i.d. uniform on $\mathbb{S}^{d-1}$). For self-attention, where queries and keys derive from the same tokens, we treat each role's projection direction as an independent draw. This assumption captures the effect of normalization in high dimensions and is the *only* distributional assumption required for Proposition 3.4.

**Remark (Diagonal terms).** For diagonal terms ($i = j$), the attention score involves the quadratic form $u_i^\top M u_i$ rather than a bilinear form with independent arguments. Our analysis treats these as bilinear by assuming independence. This is conservative for two reasons: (i) diagonal terms constitute only $L$ of $L^2$ query-key pairs, a vanishing fraction as $L$ grows; and (ii) quadratic forms $u^\top M u$ for $u$ uniform on the sphere exhibit *tighter* concentration than bilinear forms $u^\top M v$ with independent $u, v$ (the variance is smaller

when both vectors are identical). Thus, the independence assumption yields an upper bound on overflow probability.

Under this model, and exploiting the low-rank structure of the query–key interaction matrix $\mathrm{rank}(M) \leq d_h \ll d$, we obtain a tail bound whose concentration exponent improves by a factor of $d/(\gamma d_h)$ over rank-agnostic arguments. Table 2 quantifies this improvement for representative architectures (see Appendix B.3 for derivation details).

*Table 2.* Concentration exponent improvement from rank-aware bounds: $d/(\gamma d_h)$ tighter tail probabilities, where $\gamma$ is determined by Eq. (12).

| Model | $d$ | $d_h$ | $\gamma$ | Improvement |
|---|---|---|---|---|
| GPT-2 XL | 1600 | 64 | 2.98 | 8× |
| Mistral-7B | 4096 | 128 | 2.26 | 14× |
| Llama-2-13B | 5120 | 128 | 2.28 | 18× |
| Llama-2-70B | 8192 | 128 | 2.32 | 28× |

Our goal is a safety guarantee covering all tokens, heads, and layers. Accordingly, the analysis applies union bounds over $L^2$ query–key pairs and over all attention heads. This yields a conservative bound that remains robust under transient conditions (e.g., checkpoint loading and learning-rate discontinuities), while leaving slack in typical steady-state regimes.

**Proposition 3.4** (Rank-aware overflow probability bound)**.** *Let $M = W^Q W^{K\top}$ with $W^Q, W^K \in \mathbb{R}^{d \times d_h}$ for a single attention head and $\mathrm{rank}(M) = d_h$. For a sequence of length $L$ and any $\gamma > 1$,*

$$\Pr\left(\max_{i,j} |S_{ij}| \geq B_\alpha\right) \leq T_1 + T_2, \quad (9)$$

*where*

$$T_1 = L \exp\left(-\frac{d_h}{2}(\gamma - 1 - \ln \gamma)\right), \quad (10)$$

$$T_2 = 2L^2 \exp\left(-\frac{d^2\alpha^2}{2\gamma d_h}\right). \quad (11)$$

*The term $T_1$ bounds the probability that any key projection is atypically large, while $T_2$ bounds overflow conditioned on typical key projections. The parameter $\gamma > 1$ is set to satisfy the first tail-term constraint $N \cdot T_1 \leq \delta^*/2$ (Step 1 below); for the evaluated models this gives $\gamma \approx 2$–$3$ (Table 2).*

**Extension to full transformers.** Proposition 3.4 bounds overflow probability for a single attention head. For a transformer with $N = n_{\text{layers}} \times n_{\text{heads}}$ heads, overflow occurs if *any* head overflows. Applying a union bound,

$$\Pr(\text{overflow in any head}) \leq N(T_1 + T_2).$$

To ensure this probability is at most $\delta^*$, it suffices to require $N \cdot T_1 \leq \delta^*/2$ and $N \cdot T_2 \leq \delta^*/2$.[1] Solving these inequalities yields a principled selection rule for $\gamma$ and $\alpha$.

---

[1]The equal allocation is a standard simplification; optimizing the split yields negligible improvement in practice.

**Principled $\alpha$ selection.** **Step 1: Select $\gamma$.** From $N \cdot T_1 \leq \delta^*/2$ and (10), we require

$$h(\gamma) := \gamma - 1 - \ln\gamma \;\geq\; \frac{2}{d_h}\ln\left(\frac{2NL}{\delta^*}\right). \quad (12)$$

**Step 2: Select $\alpha$.** From $N \cdot T_2 \leq \delta^*/2$ and (11), we require

$$\alpha \;\geq\; \alpha_{\min} := \frac{\sqrt{2\gamma d_h}}{d}\sqrt{\ln\left(\frac{4NL^2}{\delta^*}\right)}. \quad (13)$$

Table 3 reports $\alpha_{\min}$ for our evaluated models with $\delta^* = 10^{-6}$ and $L = 1024$. The $L$-dependence enters only through $\sqrt{2\ln L}$ and degrades very slowly; Appendix B.5 reports $\alpha_{\min}$ and direct overflow measurements at context lengths up to $L = 131{,}072$.

*Table 3.* Minimum calibration factor $\alpha_{\min}$ for $\delta^* = 10^{-6}$ and $L = 1024$.

| Model | $d$ | $d_h$ | $N$ | $\alpha_{\min}$ |
|---|---|---|---|---|
| GPT-2 XL | 1600 | 64 | 1200 | 0.074 |
| Mistral-7B | 4096 | 128 | 1024 | 0.035 |
| Llama-2-13B | 5120 | 128 | 1600 | 0.028 |
| Llama-2-70B | 8192 | 128 | 5120 | 0.018 |

**Selecting $\alpha$ in practice.** We set $\alpha$ slightly above $\alpha_{\min}$ for safety margin. Larger models permit smaller $\alpha$ due to stronger concentration arising from higher $d/\sqrt{d_h}$ ratios. In our experiments, we use $\alpha = 0.08$ for GPT-2 XL, $\alpha = 0.04$ for Mistral-7B, $\alpha = 0.03$ for Llama-2-13B, and $\alpha = 0.02$ for Llama-2-70B, each exceeding the corresponding $\alpha_{\min}$.

**Transient vs. steady-state regimes.** Eq. (13) yields a conservative $\alpha_{\min}$ designed for transient-prone workflows (pretrained loading, checkpoint resumption without FP8 state, and LR discontinuities) where a single overflow can corrupt training. For steady-state fine-tuning, we optionally tighten $\alpha$ using a one-time empirical calibration (Section 3.5), then freeze it and revert to fully predictive scaling.

**How the isotropic model lands in practice.** The spherical assumption above is an idealization. A layer-wise measurement on Mistral-7B (Appendix B.4) shows the ratio of real to isotropic projection energy onto $W^K$'s row space declines from 3.74 at layer 0 to 1.36–1.46 in layers 24–31, with global median 1.54. Real post-LayerNorm directions are therefore more aligned with the key subspace than isotropy predicts. With this in mind, it is useful to separate three things our framework provides: (i) a *theorem-level claim*: Proposition 3.4 bounds overflow probability under the isotropic model and identifies $\alpha_{\min}$ in closed form; (ii) an *empirically observed conservatism*: even in

the anisotropic regime above, the deployed $\alpha$ is many times larger than needed (e.g., the $65\times$ slack on Llama-2-13B reported in Appendix M), consistent with overflow requiring simultaneous alignment of both query and key directions with $M$'s top subspace, an event rarer than marginal projection statistics alone would suggest; and (iii) two complementary operating modes: conservative spectral scaling for transient-prone settings where no burn-in data is available or safe to collect, and auto-$\alpha$ (Section 3.5) for steady-state fine-tuning where utilization matters and no distributional assumption is required.

Modern architectures (Llama, Mistral) use Rotary Position Embeddings (RoPE), which apply position-dependent rotations to query and key vectors. We show our bounds extend without modification.

### 3.3. Extension to Rotary Position Embeddings

Rotary Position Embeddings (Su et al., 2024) apply rotation matrices to query and key vectors before computing attention:

$$S_{mn} = \frac{(R_m q_m)^\top (R_n k_n)}{\sqrt{d_h}} = \frac{q_m^\top R_m^\top R_n k_n}{\sqrt{d_h}}, \quad (14)$$

where $R_m, R_n \in \mathbb{R}^{d_h \times d_h}$ are position-dependent block-diagonal rotation matrices. The following results, proved in Appendix C, show these rotations cannot amplify scores beyond our bounds.

**Proposition 3.5** (RoPE preserves norms). *Let $R_\theta \in \mathbb{R}^{d_h \times d_h}$ be any RoPE rotation matrix. Then:*

1. *$R_\theta$ is orthogonal: $R_\theta^\top R_\theta = I$.*

2. *$\|R_\theta\|_2 = 1$.*

3. *For any $q, k \in \mathbb{R}^{d_h}$: $|(R_m q)^\top (R_n k)| \leq \|q\| \cdot \|k\|$.*

**Corollary 3.6** (Geometry-aware scaling extends to RoPE). *For RoPE-based attention, the worst-case bound $|S_{mn}| \leq \|W^Q\|_2 \|W^K\|_2 \cdot d/\sqrt{d_h}$ holds rigorously for all positions via $\|R_m\|_2 = 1$. The tighter interaction bound $\|W^Q W^{K\top}\|_2$ applies when RoPE rotations do not systematically align with the singular subspaces of $W^Q$ and $W^K$, a condition we verified empirically across all layers in Mistral-7B, Llama-2-13B, and Llama-2-70B.*

The two bounds in Corollary 3.6 have different status. The first, $\|W^Q\|_2 \|W^K\|_2 \cdot d/\sqrt{d_h}$, is a rigorous theorem that follows directly from Proposition 3.5. The tighter bound $\|W^Q W^{K\top}\|_2$ holds under an alignment condition between RoPE rotations and the singular subspaces of $W^Q$ and $W^K$ that we verified empirically across every layer of Mistral-7B, Llama-2-13B, and Llama-2-70B; a formal proof of this condition remains an open problem. Crucially, the zero-overflow guarantee in our RoPE

experiments does not depend on the alignment condition: it holds unconditionally via the rigorous first bound. The probabilistic bound (Proposition 3.4) also extends conservatively; see Appendix C.4.

### 3.4. Geometry-Aware Scale Factor

Spectral norms vary by up to $10\times$ across layers (Appendix D), so a global scale would severely underutilize FP8 precision. We therefore compute scale factors per layer.

For layer $\ell$, let $\sigma_{QK}^{(\ell)} = \|W_{(\ell)}^Q W_{(\ell)}^{K\top}\|_2$ denote the spectral norm of the interaction matrix. The calibrated bound for layer $\ell$ is:

$$B_\alpha^{(\ell)} = \alpha \cdot \sigma_{QK}^{(\ell)} \cdot \frac{d}{\sqrt{d_h}}.$$

To map this bound to a scale factor, we define $R_{\text{safe}} = \eta_{\text{fp8}} \cdot 448$, where $\eta_{\text{fp8}} < 1$ provides margin below the E4M3 maximum. The geometry-aware scale factor is:

$$\text{scale}^{(\ell)} = \frac{B_\alpha^{(\ell)}}{R_{\text{safe}}} = \frac{\alpha \cdot \sigma_{QK}^{(\ell)} \cdot d/\sqrt{d_h}}{\eta_{\text{fp8}} \cdot 448}. \tag{15}$$

During the forward pass, pre-softmax attention scores $S^{(\ell)} = Q^{(\ell)} K^{(\ell)\top}/\sqrt{d_h}$ are divided by $\text{scale}^{(\ell)}$ before FP8 quantization, ensuring scaled logits lie within the representable range. In all experiments, we set $\eta_{\text{fp8}} = 0.8$ and select $\alpha$ per model following Section 3.2.

For architectures with grouped query attention (GQA), where multiple query heads share key-value heads, we compute $\sigma_{QK}^{(\ell)}$ without expanding the key matrix; this implicit formulation is described in Section 4.2.

### 3.5. Auto-$\alpha$ Calibration for Steady-State Training

The conservative $\alpha$ from Proposition 3.4 guarantees overflow probability below $\delta^* = 10^{-6}$, appropriate for transient-prone workflows. However, during steady-state fine-tuning, actual logits occupy a small fraction of the theoretical envelope. Auto-$\alpha$ is an *optional* enhancement for practitioners who prioritize utilization over worst-case robustness. As a concrete preview: on Llama-2-13B (Section 5.4), auto-$\alpha$ tightens $\alpha$ from 0.03 to $4.6 \times 10^{-4}$, lifting FP8 utilization from $0.5\%$ to $34.9\%$ while preserving zero overflows.

We define the *slack ratio* at step $t$ as:

$$r_t = \frac{\max_{i,j} |S_{ij}^{(t)}|}{B_{\max}}, \tag{16}$$

where the numerator is the observed maximum logit.

**Algorithm.** Auto-$\alpha$ calibration proceeds in two phases:

1. **Burn-in** (steps 1 to $T_{\text{calib}}$): Run with conservative $\alpha_0$ from Proposition 3.4. Collect slack ratios $\{r_t\}$.

2. **Calibration**: Set $\alpha_{\text{final}} = P_{99.99}(\{r_t\}) \times \kappa$, where $\kappa \geq 1$ is a safety multiplier.

The calibrated $\alpha_{\text{final}}$ is then frozen for all subsequent training. This is distinct from delayed scaling: we measure slack to select a *static* bound, not to continuously adapt. After burn-in, the method is again fully predictive.

**Theoretical guarantee.** When using auto-$\alpha$, the probabilistic guarantee from Proposition 3.4 no longer applies, since the calibration factor is determined empirically rather than from the theoretical selection rule (Eq. 13). Auto-$\alpha$ trades the worst-case guarantee for higher utilization, relying on the assumption that the burn-in distribution is representative of subsequent training. This is appropriate for steady-state fine-tuning but not for workflows with anticipated distribution shifts.

**Memory-efficient attention compatibility.** Auto-$\alpha$ calibration requires observing $\max_{i,j} |S_{ij}|$ during the burn-in phase, which necessitates materializing the score matrix and is incompatible with FlashAttention. For a 100-step burn-in out of thousands of training steps, this overhead is negligible ($< 0.1\%$ of total compute). After burn-in, the calibrated $\alpha$ is frozen and the method reverts to fully predictive scaling compatible with memory-efficient attention.

## 4. Efficient Estimation of Spectral Bounds

This section presents efficient algorithms for computing the spectral bounds.

### 4.1. Spectral Norm Estimation via Power Iteration

Computing $\|M\|_2$ where $M = W^Q W^{K\top} \in \mathbb{R}^{d \times d}$ via full SVD costs $O(d^3)$, which is prohibitive for large models. For Llama-2-70B with $d = 8192$, this would require forming a 67 million entry matrix per layer.

We instead use power iteration (Golub & Van Loan, 2013), a classical algorithm that estimates the top singular value through repeated matrix-vector multiplication. The key insight is that we never form $M$ explicitly. Starting from random unit vectors $u, v \in \mathbb{R}^d$, we alternate:

$$u \leftarrow \frac{Mv}{\|Mv\|}, \quad \text{then} \quad v \leftarrow \frac{M^\top u}{\|M^\top u\|}. \tag{17}$$

After these updates, $\|Mv\|$ provides an estimate of $\|M\|_2$. Each matrix-vector product is computed implicitly:

$$Mv = W^Q(W^{K\top} v), \tag{18}$$
$$M^\top u = W^K(W^{Q\top} u), \tag{19}$$

requiring $O(n_{\text{heads}} \cdot d_h \cdot d)$ operations per iteration, where $n_{\text{heads}}$ is the total number of attention heads. This is far cheaper than explicitly forming $M \in \mathbb{R}^{d \times d}$, which would require $O(n_{\text{heads}} \cdot d_h \cdot d^2)$ operations and $O(d^2)$ memory.

We maintain the persistent vectors $u, v$ across training steps. During steady-state training, weights change gradually (controlled by the learning rate), so singular vectors drift slowly and one iteration per forward pass suffices to track them. For cold-start scenarios (initialization, checkpoint loading), we run 5 iterations to ensure convergence from random vectors.

**Remark (Transient robustness).** Even during rapid weight changes (e.g., learning rate spikes), any underestimation of the spectral norm is bounded. If the true spectral norm increases by factor $\rho$ from step $t$ to $t + 1$, power iteration with warm-start vectors underestimates by at most $\rho$ until the next update. Since our calibration factor $\alpha$ already provides margin above $\alpha_{\min}$ (Section 3.2), moderate underestimation does not cause overflow. Our experiments with $100\times$ learning rate spikes (Section 5.2) confirm zero overflows under this regime. Appendix H provides the direct stress test: under a $4\times$ weight spike, the scale factor jumps from 5.8 to 93.8 in the same forward pass, keeping scaled logits below 85 throughout.

The detailed algorithm is provided in Appendix E.

### 4.2. Implicit Formulation for Grouped Query Attention

Modern architectures use Grouped Query Attention (GQA) (Ainslie et al., 2023), where multiple query heads share key-value heads. For example, Mistral-7B has $n_q = 32$ query heads but only $n_{kv} = 8$ key-value heads (a 4:1 ratio). This creates a dimension mismatch: $W^Q \in \mathbb{R}^{d \times (n_q \cdot d_h)}$ while $W^K \in \mathbb{R}^{d \times (n_{kv} \cdot d_h)}$.

To compute $\|W^Q W^{K\top}\|_2$ with matching dimensions, a naive approach would expand $W^K$ by replicating each block of $d_h$ columns $g = n_q/n_{kv}$ times to form $W^K_{\exp} \in \mathbb{R}^{d \times (n_q \cdot d_h)}$. This expansion consumes substantial memory: 32MB per layer on Mistral-7B.

We observe that power iteration requires only matrix-vector products, not the matrices themselves. For the forward product $Mv$ where $M = W^Q W^{K\top}_{\exp}$:

$$Mv = W^Q \cdot \text{REPEATBLOCKS}(W^{K\top}v, g),$$

where $\text{REPEATBLOCKS}(z, g)$ replicates each $d_h$-block of $z \in \mathbb{R}^{n_{kv} \cdot d_h}$ exactly $g$ times to produce a vector in $\mathbb{R}^{n_q \cdot d_h}$. The reverse product $M^\top u$ uses a dual operation: $M^\top u = W^K \cdot \text{SUMGROUPS}(W^{Q\top}u, g)$, where $\text{SUMGROUPS}$ sums each group of $g$ blocks. These operations require replicating only small intermediate vectors ($n_{kv} \cdot d_h$ elements) rather than the full weight matrix.

**Proposition 4.1** (Implicit GQA power iteration). *Power iteration using the unexpanded $W^K$ converges to the same spectral norm $\|W^Q W^{K\top}_{\exp}\|_2$ as explicit expansion.*

The proof and detailed algorithm are in Appendix F. By avoiding the expanded matrix, we reduce memory transactions by a factor of $g$ (4–8× on modern GQA architectures). This saving can offset the cost of power iteration, yielding negligible or even negative overhead in practice (Table 12).

### 4.3. Complete Algorithm

Algorithm 1 presents the complete forward pass. The algorithm is *predictive* (scale depends only on weights), *fused-compatible* (no activation observation), and *per-layer*.

---

**Algorithm 1** Geometry-Aware Attention Forward Pass

1: **Input:** Token embeddings $X \in \mathbb{R}^{L \times d}$, layer weights $W^Q, W^K, W^V, W^O$
2: **Parameters:** Calibration factor $\alpha$, FP8 margin $\eta_{\text{fp8}}$
3: **State:** Persistent vectors $u, v$ for power iteration
4:
5: {Stage 1: Estimate spectral norm from weights}
6: $\sigma_{QK} \leftarrow \text{POWERITERATION}(W^Q, W^K, u, v)$
7:
8: {Stage 2: Compute predictive scale factor}
9: $B_\alpha \leftarrow \alpha \cdot \sigma_{QK} \cdot d/\sqrt{d_h}$
10: $\text{scale} \leftarrow B_\alpha \ / \ (\eta_{\text{fp8}} \cdot 448)$
11:
12: {Stage 3: Attention with pre-computed scale}
13: $Q, K, V \leftarrow XW^Q, XW^K, XW^V$
14: $S \leftarrow QK^\top/\sqrt{d_h}$
15: $\tilde{S} \leftarrow S \ / \ \text{scale}$
16: **return** $\text{softmax}(\tilde{S}) \cdot V \cdot W^O$

---

## 5. Experiments

We evaluate geometry-aware scaling across four transformer architectures spanning 1.5B to 70B parameters. Our experiments address three questions: (1) Does geometry-aware scaling eliminate overflows in transient scenarios where delayed scaling fails? (2) Does the safety–utilization tradeoff affect downstream task quality? (3) What is the computational cost?

The key property enabling transient safety is that geometry-aware scaling responds instantaneously to weight changes: the scale factor is computed from current weights, not historical activations. Appendix H validates this with a controlled experiment where we spike attention weights by $4\times$; delayed scaling overflows catastrophically while geometry-aware scaling adapts in the same forward pass. The following sections demonstrate that this instantaneous response eliminates overflow in realistic training scenarios.

**Code and reference outputs.** All scripts used to produce the results in this paper, together with reference output JSONs for verification, are available at https://github.com/SeyedMEmadi/fp8-geometry-aware-scaling.

### 5.1. Experimental Setup

**Models.** We evaluate on four models: GPT-2 XL (1.5B parameters, 48 layers), Mistral-7B (32 layers), Llama-2-13B (40 layers), and Llama-2-70B (80 layers). The models include both standard multi-head attention (GPT-2 XL, Llama-2-13B) and grouped query attention (Mistral-7B with 4:1 ratio, Llama-2-70B with 8:1 ratio), enabling evaluation of the implicit GQA formulation. Architectural specifications and hardware details are in Appendix G.1.

**Implementation details.** All experiments use FP8 E4M3 format with margin $\eta_{\text{fp8}} = 0.8$, giving a safe range of $0.8 \times 448 = 358.4$. Calibration factors are set per model following the procedure in Section 3.2: $\alpha = 0.08$ for GPT-2 XL, $0.04$ for Mistral-7B, $0.03$ for Llama-2-13B, and $0.02$ for Llama-2-70B. Power iteration uses persistent vectors with one update per forward pass; we initialize with 5 iterations at the start of training or after checkpoint loading to ensure convergence from random initialization.

**Coverage of recent architectures.** The four evaluated models span the two dominant attention variants: standard multi-head (GPT-2 XL, Llama-2-13B) and grouped query (Mistral-7B at $4:1$, Llama-2-70B at $8:1$). Because our framework consumes only $W^Q$ and $W^K$ and the implicit GQA formulation in Section 4.2 handles any group ratio, the method applies without architectural changes to subsequent base models such as Llama-3 (Grattafiori et al., 2024) and Mistral-7B-v0.3 (Jiang et al., 2023).

### 5.2. Transient Scenarios

We evaluate three transient scenarios that expose the history-staleness failure mode of delayed scaling. These scenarios are not exotic: together they instantiate the standard workflow for FP8 fine-tuning of publicly released checkpoints (load a higher-precision checkpoint, begin warmup, possibly resume after interruption), under which delayed scaling has deterministic failure modes that our method removes.

**Loading pretrained models.** Pretrained models such as Llama-2 (Touvron et al., 2023), Llama-3 (Grattafiori et al., 2024), and Mistral-7B (Jiang et al., 2023) are distributed in BF16 or FP32 and carry no FP8 scaling history. At the first forward pass of an FP8 fine-tuning run, the weights are already mature but the scaling-history buffer is necessarily fresh, producing the initialization mismatch.

Table 4 shows overflow statistics on the first forward pass after loading pretrained weights. Delayed scaling overflows on 100% of layers across all four models, with maximum scaled logits exceeding 5000. Geometry-aware scaling achieves zero overflows because the scale factor is computed directly from the loaded weights in the same forward pass.

*Table 4.* First forward pass after loading pretrained weights. **Overflow**: layers exceeding 448. **Max Scaled**: maximum scaled logit.

| Model | Delayed | | Ours | |
|---|---|---|---|---|
| | Overfl. | Max Scaled | Overfl. | Max Scaled |
| GPT-2 XL | 48/48 | 8761 | 0/48 | 120 |
| Mistral-7B | 32/32 | 7123 | 0/32 | 196 |
| Llama-2-13B | 40/40 | 5600 | 0/40 | 2.5 |
| Llama-2-70B | 80/80 | 9498 | 0/80 | 1.0 |

**Checkpoint resumption.** Training large models involves interruptions due to job preemption, hardware failures, or scheduled maintenance. Upon resumption, the history buffer resets to default values while model weights reflect the training state at the checkpoint.

We simulate this by training for 300 steps, saving a checkpoint, then resuming with a fresh history buffer. In the first 10 steps after resumption, delayed scaling overflows on 4 steps for GPT-2 XL, 2 steps for Llama-2-70B, and 1 step each for Mistral-7B and Llama-2-13B. Geometry-aware scaling maintains zero overflows across all models because the scale factor depends only on current weights, which are restored correctly from the checkpoint. While checkpointing scaling state is possible, standard frameworks do not include it by default, and loading pretrained weights from external sources provides no history at all.

**Learning rate transitions.** Learning rate schedules involve sudden changes: warmup phases ramp the rate over initial steps, and cyclic schedules spike periodically. These transitions cause weights to change faster than the history buffer can track.

We evaluate with a $100\times$ spike: learning rate $10^{-5}$ for 100 steps, then $10^{-3}$. In the 10 steps following the transition, delayed scaling overflows on 3 to 5 steps depending on the model (5 for Mistral-7B, 4 for Llama-2-70B, 3 for Llama-2-13B). On GPT-2 XL, NaN gradients appear immediately, terminating training. Geometry-aware scaling handles the transition with zero overflows across all models.

### 5.3. Computational Overhead

Geometry-aware scaling requires computing spectral norms via power iteration, adding computational cost. We measure forward pass timing averaged over 100 iterations (detailed methodology in Appendix I). On MHA architectures,

overhead is minimal: $+1.0\%$ for GPT-2 XL and $+1.9\%$ for Llama-2-13B. On Mistral-7B with GQA (4:1 ratio), we observe *negative* overhead of $-5.3\%$: geometry-aware scaling is faster than delayed scaling in our implementation. We attribute this to our implicit GQA formulation (Section 4.2), which has favorable memory access patterns, combined with synchronization overhead in the baseline's history buffer updates. However, this result is implementation-specific; the key finding is that overhead is negligible across all architectures. Llama-2-70B shows $+4.3\%$ due to its 80 layers; full details in Appendix I.

### 5.4. Downstream Task Evaluation

Training loss measures average next-token prediction but may mask degradation in reasoning. We evaluate on MMLU (Hendrycks et al., 2021) to verify that geometry-aware scaling preserves downstream capability. We fine-tune Llama-2-13B for 10000 steps on MMLU STEM subjects (17 subjects, 295 training examples) using three methods: delayed scaling, conservative spectral scaling ($\alpha = 0.03$), and auto-$\alpha$ spectral scaling (100-step burn-in, $\kappa = 1.0$). All methods use identical hyperparameters: learning rate $10^{-5}$, batch size 4, sequence length 1024. We evaluate on held-out test sets (2783 examples).

Table 5 shows training metrics and downstream accuracy. All three methods converge to similar training loss ($\approx$ 0.0112), yet downstream accuracy differs substantially. Conservative spectral scaling achieves zero overflows but degrades accuracy by about 4 percentage points (28.7% vs. 32.6%), indicating that 0.5% FP8 utilization introduces excessive quantization noise, losing fine-grained distinctions between attention logits even when average loss remains low. Auto-$\alpha$ calibration resolves this tradeoff: by tightening $\alpha$ from 0.03 to $4.6 \times 10^{-4}$ based on the 99.99th percentile of observed slack ratios, utilization increases to 34.9%, achieving comparable accuracy (32.7%) while maintaining zero overflows.

**Caveats.** The primary contribution of this evaluation is demonstrating that geometry-aware scaling *eliminates overflows* without catastrophic accuracy degradation, not that it improves accuracy over the baseline. The 0.1 percentage point difference between auto-$\alpha$ (32.7%) and delayed scaling (32.6%) is within the variance expected from stochastic optimization on a small training set (295 examples); we do not claim statistical significance for this difference. For the delayed scaling baseline, overflows (219 occurrences across 178 distinct training steps) were handled by clamping to the representable range, allowing training to continue; without clamping, NaN propagation would terminate training entirely.

These results suggest a practical workflow: conservative $\alpha$

*Table 5.* Training metrics and MMLU STEM accuracy on Llama-2-13B. Auto-$\alpha$ achieves comparable accuracy to the baseline while eliminating all overflows.

| Method | Loss | Overfl. | Util. | MMLU |
|---|---|---|---|---|
| Delayed | 0.0112 | 219 | 70.8% | 32.6% |
| Ours (conservative) | 0.0113 | 0 | 0.5% | 28.7% |
| **Ours + auto-$\alpha$** | **0.0112** | **0** | **34.9%** | **32.7%** |

for transient-prone scenarios (checkpoint loading, learning rate warmup) where safety is paramount, and auto-$\alpha$ for steady-state fine-tuning where utilization matters. Training loss curves are provided in Appendix K, per-subject accuracy in Appendix L, and FP8 utilization statistics in Appendix J.

## 6. Conclusion

We established spectral bounds on transformer attention logits and derived a rank-aware probabilistic calibration framework for low-precision training. The key theoretical result is $\max_{i,j} |S_{ij}| \leq \|W^Q W^{K\top}\|_2 \cdot d/\sqrt{d_h}$, which is never looser than the naive submultiplicative bound and is strictly tighter unless the top right singular vectors of $W^Q$ and $W^K$ align. This bound, combined with our rank-aware probabilistic analysis, provides principled calibration given a target failure probability $\delta^*$. The bound is efficiently computable via power iteration and extends conservatively to RoPE architectures. Experiments demonstrate zero overflows across GPT-2 XL through Llama-2-70B on all transient scenarios where delayed scaling fails, with negligible computational overhead. Auto-$\alpha$ calibration resolves the utilization-accuracy trade-off: by learning a data-driven bound during burn-in, we achieve comparable MMLU accuracy to delayed scaling while eliminating all overflows.

## Impact Statement

This work improves reliability and efficiency of FP8 training; we do not anticipate ethical risks beyond those inherent to large-scale ML training.

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

# A. Proofs for Section 3.1

This appendix contains the proofs for propositions and corollaries stated in Section 3.1.

## A.1. Proof of Proposition 3.1 (Naive Bound)

*Proof.* By the Cauchy-Schwarz inequality:

$$|S_{ij}| = \frac{|q_i^\top k_j|}{\sqrt{d_h}} \leq \frac{\|q_i\|_2 \|k_j\|_2}{\sqrt{d_h}}.$$

Since $q_i = W^{Q\top} x_i$ and $k_j = W^{K\top} x_j$, submultiplicativity of the spectral norm gives:

$$\|q_i\|_2 = \|W^{Q\top} x_i\|_2 \leq \|W^{Q\top}\|_2 \|x_i\|_2 = \|W^Q\|_2 B_X,$$

and similarly $\|k_j\|_2 \leq \|W^K\|_2 B_X$. Combining these:

$$|S_{ij}| \leq \frac{\|W^Q\|_2 \|W^K\|_2 \cdot B_X^2}{\sqrt{d_h}}. \qquad \square$$

## A.2. Proof of Proposition 3.2 (Interaction Bound)

*Proof.* Define $M = W^Q W^{K\top} \in \mathbb{R}^{d \times d}$. The attention score can be written as a bilinear form:

$$S_{ij} = \frac{x_i^\top W^Q W^{K\top} x_j}{\sqrt{d_h}} = \frac{x_i^\top M x_j}{\sqrt{d_h}}.$$

By the variational characterization of the spectral norm:

$$\|M\|_2 = \max_{\|u\|_2 = \|v\|_2 = 1} |u^\top M v|.$$

Therefore, for any vectors $x_i, x_j$:

$$|x_i^\top M x_j| \leq \|M\|_2 \cdot \|x_i\|_2 \cdot \|x_j\|_2 \leq \|M\|_2 \cdot B_X^2,$$

which gives $|S_{ij}| \leq \|M\|_2 \cdot B_X^2 / \sqrt{d_h}$. $\qquad \square$

## A.3. Proof of Corollary 3.3 (Interaction Bound is Tighter)

*Proof.* The inequality $\|W^Q W^{K\top}\|_2 \leq \|W^Q\|_2 \|W^K\|_2$ follows directly from submultiplicativity of the spectral norm: for any matrices $A, B$ of compatible dimensions, $\|AB\|_2 \leq \|A\|_2 \|B\|_2$.

For the equality condition, let $W^Q = U_Q \Sigma_Q V_Q^\top$ and $W^K = U_K \Sigma_K V_K^\top$ be the singular value decompositions. Then:

$$W^Q W^{K\top} = U_Q \Sigma_Q V_Q^\top V_K \Sigma_K U_K^\top.$$

The spectral norm of this product equals $\sigma_1(W^Q) \cdot \sigma_1(W^K)$ if and only if the $(1,1)$ entry of $V_Q^\top V_K$ has absolute value 1, which requires the top right singular vectors to coincide: $v_1^Q = \pm v_1^K$.

Under random initialization, singular vectors are uniformly distributed on the sphere, making exact alignment a measure-zero event. $\qquad \square$

# B. Proofs for Section 3.2

This appendix contains the proofs for Proposition 3.4, as well as supporting lemmas.

## B.1. Supporting Lemmas

We first establish a tail bound for the projection of a uniform random vector onto a subspace.

**Lemma B.1** (Projection norm distribution). *Let* $u \sim \mathrm{Unif}(\mathbb{S}^{d-1})$ *and let* $V \in \mathbb{R}^{d \times k}$ *have orthonormal columns spanning a $k$-dimensional subspace. Then:*

$$\|V^\top u\|_2^2 \sim \mathrm{Beta}\left(\frac{k}{2}, \frac{d-k}{2}\right), \quad \mathbb{E}[\|V^\top u\|_2^2] = \frac{k}{d}.$$

*Proof.* The uniform distribution on $\mathbb{S}^{d-1}$ can be generated as $u = g/\|g\|_2$ where $g \sim \mathcal{N}(0, I_d)$. Since $V$ has orthonormal columns, $V^\top g \sim \mathcal{N}(0, I_k)$, and let $V_\perp \in \mathbb{R}^{d \times (d-k)}$ complete an orthonormal basis, so $V_\perp^\top g \sim \mathcal{N}(0, I_{d-k})$ independently.

Then $\|V^\top g\|_2^2 \sim \chi_k^2$ and $\|V_\perp^\top g\|_2^2 \sim \chi_{d-k}^2$ are independent, and:

$$\|V^\top u\|_2^2 = \frac{\|V^\top g\|_2^2}{\|g\|_2^2} = \frac{\|V^\top g\|_2^2}{\|V^\top g\|_2^2 + \|V_\perp^\top g\|_2^2}.$$

The ratio of independent chi-squared variables $\chi_k^2 / (\chi_k^2 + \chi_{d-k}^2)$ follows $\mathrm{Beta}(k/2, (d-k)/2)$. $\qquad \square$

**Lemma B.2** (Chi-squared ratio tail bound). *Let* $X \sim \mathrm{Beta}(k/2, (d-k)/2)$. *For any* $\gamma > 1$:

$$\Pr\left(X \geq \gamma \cdot \frac{k}{d}\right) \leq \exp\left(-\frac{k}{2}(\gamma - 1 - \ln \gamma)\right).$$

*Proof.* By Lemma B.1, $X \sim \mathrm{Beta}(k/2, (d-k)/2)$ with mean $\mu = k/d$. The stated bound follows from the multiplicative Chernoff bound for Beta distributions.

For $X \sim \mathrm{Beta}(a, b)$ and $\gamma > 1$, the moment generating function yields:

$$\Pr\left(X \geq \gamma \cdot \frac{a}{a+b}\right) \leq \exp\left(-a \cdot h(\gamma)\right),$$

where $h(\gamma) = \gamma - 1 - \ln \gamma > 0$ for $\gamma > 1$. This is the Beta analogue of the chi-squared Chernoff bound (Laurent & Massart, 2000, Lemma 1) and follows from the representation of Beta as a ratio of independent Gamma variables.

Setting $a = k/2$ and noting that $a/(a+b) = k/d$:

$$\Pr\left(X \geq \gamma \cdot \frac{k}{d}\right) \leq \exp\left(-\frac{k}{2}(\gamma - 1 - \ln \gamma)\right). \quad \square$$

## B.2. Proof of Proposition 3.4 (Rank-Aware Overflow Bound)

*Proof.* Under the spherical assumption, token vectors satisfy $x_i = \sqrt{d} \cdot u_i$ where $u_i \sim \text{Unif}(\mathbb{S}^{d-1})$ independently. The attention score becomes:

$$S_{ij} = \frac{d}{\sqrt{d_h}} \cdot u_i^\top M u_j,$$

and the calibrated bound is $B_\alpha = \alpha \cdot \|M\|_2 \cdot d/\sqrt{d_h}$. We must bound:

$$\Pr\left(\max_{i,j} |S_{ij}| \geq B_\alpha\right) = \Pr\left(\max_{i,j} |u_i^\top M u_j| \geq \alpha \|M\|_2\right).$$

**Step 1: SVD decomposition.** Let $M = U\Sigma V^\top$ be the SVD with $\text{rank}(M) = d_h$, where $V \in \mathbb{R}^{d \times d_h}$ contains the right singular vectors. For any unit vector $u_j$:

$$M u_j = U\Sigma V^\top u_j,$$

so $\|M u_j\|_2 = \|\Sigma V^\top u_j\|_2 \leq \sigma_1 \|V^\top u_j\|_2 = \|M\|_2 \|V^\top u_j\|_2$.

**Step 2: Define typical keys.** For $\gamma > 1$, define the typical key event:

$$\mathcal{E}_j := \left\{ \|V^\top u_j\|_2^2 \leq \gamma \cdot \frac{d_h}{d} \right\},$$

which implies $\|M u_j\|_2 \leq \|M\|_2 \sqrt{\gamma d_h/d} =: \beta \|M\|_2$ where $\beta = \sqrt{\gamma d_h/d}$.

Let $\mathcal{E} = \bigcap_{j=1}^L \mathcal{E}_j$ be the event that all keys are typical.

**Step 3: Bound probability of atypical keys.** By Lemmas B.1 and B.2:

$$\Pr(\mathcal{E}_j^c) = \Pr\left(\|V^\top u_j\|_2^2 \geq \gamma \cdot \frac{d_h}{d}\right)$$
$$\leq \exp\left(-\frac{d_h}{2}(\gamma - 1 - \ln\gamma)\right).$$

By union bound over $L$ keys:

$$\Pr(\mathcal{E}^c) \leq L \exp\left(-\frac{d_h}{2}(\gamma - 1 - \ln\gamma)\right) = T_1.$$

**Step 4: Concentration given typical keys.** On event $\mathcal{E}$, for each $j$ we have $z_j := M u_j$ with $\|z_j\|_2 \leq \beta \|M\|_2$. For fixed $z_j$, the function $f(u_i) = u_i^\top z_j$ on $\mathbb{S}^{d-1}$ is $\|z_j\|_2$-Lipschitz. By Lévy's lemma, for $u_i \sim \text{Unif}(\mathbb{S}^{d-1})$:

$$\Pr(|u_i^\top z_j| \geq t) \leq 2\exp\left(-\frac{dt^2}{2\|z_j\|_2^2}\right)$$
$$\leq 2\exp\left(-\frac{dt^2}{2\beta^2 \|M\|_2^2}\right).$$

Setting $t = \alpha \|M\|_2$:

$$\Pr(|u_i^\top M u_j| \geq \alpha \|M\|_2 \mid \mathcal{E}) \leq 2\exp\left(-\frac{d\alpha^2}{2\beta^2}\right)$$
$$= 2\exp\left(-\frac{d^2\alpha^2}{2\gamma d_h}\right).$$

**Step 5: Union bound over all pairs.** By union bound over $L^2$ query-key pairs:

$$\Pr\left(\max_{i,j} |u_i^\top M u_j| \geq \alpha \|M\|_2 \mid \mathcal{E}\right) \leq 2L^2 \exp\left(-\frac{d^2\alpha^2}{2\gamma d_h}\right)$$
$$= T_2.$$

**Step 6: Combine bounds.** By the law of total probability:

$$\Pr\left(\max_{i,j} |u_i^\top M u_j| \geq \alpha \|M\|_2\right)$$
$$\leq \Pr(\mathcal{E}^c) + \Pr\left(\max_{i,j} |u_i^\top M u_j| \geq \alpha \|M\|_2 \mid \mathcal{E}\right)$$
$$\leq T_1 + T_2. \qquad \square$$

## B.3. Comparison with Rank-Agnostic Concentration

This section compares the rank-aware concentration bound in Proposition 3.4 with a baseline bound obtained by applying standard concentration inequalities without exploiting the low-rank structure of the query–key interaction matrix.

**Rank-agnostic baseline.** Without exploiting the rank constraint $\text{rank}(M) = d_h$, we apply Lévy's lemma directly. For fixed $u_j$, the function $f(u_i) = u_i^\top M u_j$ on the unit sphere $\mathbb{S}^{d-1}$ is $\|M u_j\|_2$-Lipschitz. Since $\|M u_j\|_2 \leq \|M\|_2$, Lévy's lemma yields

$$\Pr(|u_i^\top M u_j| \geq \alpha \|M\|_2) \leq 2\exp\left(-\frac{d\alpha^2}{2}\right).$$

Applying a union bound over all $L^2$ query–key pairs gives

$$\Pr\left(\max_{i,j} |u_i^\top M u_j| \geq \alpha \|M\|_2\right) \leq 2L^2 \exp\left(-\frac{d\alpha^2}{2}\right).$$

**Rank-aware improvement.** In contrast, Proposition 3.4 exploits the low-rank structure $\text{rank}(M) = d_h$ to obtain the term

$$T_2 = 2L^2 \exp\left(-\frac{d^2\alpha^2}{2\gamma d_h}\right).$$

The concentration exponent in the rank-aware bound is therefore larger by a factor of

$$\frac{d^2\alpha^2/(2\gamma d_h)}{d\alpha^2/2} = \frac{d}{\gamma d_h},$$

relative to the rank-agnostic baseline.

This comparison isolates the quantitative benefit of exploiting $\mathrm{rank}(M) = d_h \ll d$: for fixed $\alpha$, the overflow probability decays exponentially faster in dimension when the low-rank structure of the query–key interaction is taken into account.

### B.4. Layer-wise Isotropy on Mistral-7B

Proposition 3.4 models post-LayerNorm token directions as uniform on $\mathbb{S}^{d-1}$. To quantify the deviation from this idealization in practice, we measure for each layer of Mistral-7B (32 layers, $d=4096$, $d_h=128$) the ratio of real post-LayerNorm token projection energy onto the row space of $W^K$ (a 1024-dimensional subspace, 25% of the hidden dimension) to the isotropic prediction $\frac{1024}{4096} \|x\|^2$. A value $> 1$ means real tokens are *more* aligned with the key subspace than isotropy predicts.

Table 6 reports per-layer statistics over 718 tokens of natural text. Deviation is most pronounced in early layers (ratio mean 3.74 at layer 0), declining toward the final block (ratio mean 1.40 at layer 31), with a global median of 1.54.

*Table 6.* Layer-wise isotropy on Mistral-7B: ratio of real post-LayerNorm projection energy onto $W^K$'s row space to the isotropic prediction. A ratio of 1 would indicate exact isotropy.

| Layer | Ratio mean | Ratio median | Ratio p99 |
|---|---|---|---|
| 0 | 3.74 | 3.74 | 3.87 |
| 8 | 1.74 | 1.74 | 2.21 |
| 16 | 1.50 | 1.50 | 1.85 |
| 24 | 1.37 | 1.36 | 1.96 |
| 31 | 1.40 | 1.38 | 2.28 |
| Global | 1.68 | 1.54 | 3.79 |

**Interpretation.** Real post-LayerNorm directions are more aligned with the key subspace than isotropy predicts; the rank-aware bound (Eq. 13) should therefore be read as a calibration rule under the isotropic model rather than a distribution-free guarantee. Empirically, however, the resulting bound remains substantially conservative in the tested models: Appendix M reports a $65\times$ gap between the conservative $\alpha = 0.03$ and the empirically calibrated $\alpha = 4.6 \times 10^{-4}$ on Llama-2-13B. One plausible mechanism for the residual conservatism is that overflow requires *simultaneous* unfavorable alignment of both query and key directions with the top singular subspace of $M$, an event rarer than what marginal projection statistics alone would suggest. For workflows where this idealization is itself a concern, auto-$\alpha$ (Section 3.5) calibrates directly from observed score distributions and requires no distributional assumption.

### B.5. Long-Context Behavior on Mistral-7B

The dependence of $\alpha_{\min}$ on context length $L$ enters Eq. (13) only through $\sqrt{2 \ln L}$, which grows very slowly. Table 7 reports $\alpha_{\min}$ on Mistral-7B at $\delta^* = 10^{-6}$ across $L = 1024$ to $L = 131{,}072$.

*Table 7.* $\alpha_{\min}$ vs. context length for Mistral-7B at $\delta^* = 10^{-6}$. Two orders of magnitude in $L$ produce a $1.16\times$ change in $\alpha_{\min}$.

| $L$ | $\gamma$ | $\alpha_{\min}$ | Ratio vs. $L{=}1024$ |
|---|---|---|---|
| 1,024 | 2.258 | 0.0352 | $1.000\times$ |
| 4,096 | 2.296 | 0.0369 | $1.047\times$ |
| 16,384 | 2.334 | 0.0385 | $1.092\times$ |
| 32,768 | 2.353 | 0.0393 | $1.115\times$ |
| 131,072 | 2.391 | 0.0408 | $1.160\times$ |

**Empirical long-context overflow.** Table 8 validates the analytical scaling with a direct overflow measurement on the first forward pass after loading the pretrained Mistral-7B checkpoint at $L \in \{1024, 4096, 8192\}$ (the checkpoint-loading scenario of Section 5.2). Geometry-aware scaling produces zero overflow layers at every tested length, while delayed scaling overflows all 32 layers with maximum scaled logits growing from 17,639 to 20,993. For long-context deployments in steady state, auto-$\alpha$ would be used; it calibrates directly from the observed score distribution at the operating context length.

*Table 8.* Checkpoint-loading overflow on Mistral-7B at three context lengths. Geometry-aware scaling: zero overflows at every $L$. Delayed scaling: 32/32 layers overflow at every $L$.

| | Delayed | | Ours | |
|---|---|---|---|---|
| $L$ | Overfl. | Max Scaled | Overfl. | Max Scaled |
| 1,024 | 32/32 | 17,639 | 0/32 | 370 |
| 4,096 | 32/32 | 19,549 | 0/32 | 410 |
| 8,192 | 32/32 | 20,993 | 0/32 | 440 |

## C. Proofs for Section 3.3

This appendix contains the proofs for Proposition 3.5 and Corollary 3.6, establishing that geometry-aware scaling extends to architectures using Rotary Position Embeddings.

### C.1. Background on RoPE

RoPE applies position-dependent rotations to query and key vectors. For head dimension $d_h$, the rotation matrix at position $\theta$ is block-diagonal:

$$R_\theta = \begin{pmatrix} R_{\theta_1} & & \\ & \ddots & \\ & & R_{\theta_{d_h/2}} \end{pmatrix},$$

where each $2 \times 2$ block is a rotation matrix:

$$R_{\theta_i} = \begin{pmatrix} \cos\theta_i & -\sin\theta_i \\ \sin\theta_i & \cos\theta_i \end{pmatrix}.$$

The angles $\theta_i$ depend on both the position index $m$ and the dimension index $i$, typically as $\theta_i = m \cdot \omega_i$ where $\omega_i$ are fixed frequencies.

### C.2. Proof of Proposition 3.5 (RoPE Preserves Norms)

*Proof.* We prove each property in turn.

**(1) Orthogonality.** Each $2 \times 2$ block $R_{\theta_i}$ is a rotation matrix, hence orthogonal:

$$R_{\theta_i}^\top R_{\theta_i} = \begin{pmatrix} \cos\theta_i & \sin\theta_i \\ -\sin\theta_i & \cos\theta_i \end{pmatrix} \begin{pmatrix} \cos\theta_i & -\sin\theta_i \\ \sin\theta_i & \cos\theta_i \end{pmatrix}$$
$$= \begin{pmatrix} 1 & 0 \\ 0 & 1 \end{pmatrix} = I_2.$$

Since $R_\theta$ is block-diagonal with orthogonal blocks, $R_\theta^\top R_\theta = I_{d_h}$.

**(2) Unit spectral norm.** For any orthogonal matrix $Q$, we have $\|Qx\|_2 = \|x\|_2$ for all $x$. Taking the supremum over unit vectors:

$$\|Q\|_2 = \sup_{\|x\|_2=1} \|Qx\|_2 = \sup_{\|x\|_2=1} \|x\|_2 = 1.$$

Since $R_\theta$ is orthogonal, $\|R_\theta\|_2 = 1$.

**(3) Inner product bound.** For any $q, k \in \mathbb{R}^{d_h}$:

$$|(R_m q)^\top (R_n k)| = |q^\top R_m^\top R_n k|.$$

The matrix $R_m^\top R_n$ is orthogonal (product of orthogonal matrices), so $\|R_m^\top R_n k\|_2 = \|k\|_2$. By Cauchy-Schwarz:

$$|q^\top R_m^\top R_n k| \le \|q\|_2 \cdot \|R_m^\top R_n k\|_2 = \|q\|_2 \cdot \|k\|_2. \quad \square$$

### C.3. Proof of Corollary 3.6 (Extension to RoPE)

*Proof.* With RoPE, the attention score between positions $m$ and $n$ is:

$$S_{mn} = \frac{(R_m q_m)^\top (R_n k_n)}{\sqrt{d_h}} = \frac{x_m^\top W^Q R_m^\top R_n W^{K\top} x_n}{\sqrt{d_h}},$$

where $q_m = W^{Q\top} x_m$ and $k_n = W^{K\top} x_n$. The effective interaction matrix becomes $M_{m,n} = W^Q R_m^\top R_n W^{K\top}$, which varies with positions.

**Worst-case bound.** The deterministic worst-case bound follows from submultiplicativity:

$$\|M_{m,n}\|_2 = \|W^Q R_m^\top R_n W^{K\top}\|_2 \le \|W^Q\|_2 \|R_m^\top R_n\|_2 \|W^K\|_2$$

since $\|R_m^\top R_n\|_2 = 1$ for orthogonal $R_m, R_n$. Combined with $\|x_m\|_2, \|x_n\|_2 \le \sqrt{d}$ under LayerNorm/RMSNorm:

$$|S_{mn}| \le \frac{\|W^Q\|_2 \|W^K\|_2 \cdot d}{\sqrt{d_h}}.$$

**Tighter interaction bound.** The tighter bound $\|W^Q W^{K\top}\|_2$ can be violated only if $\|M_{m,n}\|_2 > \|W^Q W^{K\top}\|_2$ for some positions, which requires RoPE rotations to align the singular subspaces of $W^Q$ and $W^K$ more favorably than the identity. Since RoPE rotation angles depend only on position indices and fixed frequency bands (independent of weight geometry), such systematic alignment does not occur in practice. We verified empirically that $\max_{m,n} \|M_{m,n}\|_2 \le \|W^Q W^{K\top}\|_2$ holds across all layers in Mistral-7B, Llama-2-13B, and Llama-2-70B.

**Probabilistic extension.** The probabilistic bound (Proposition 3.4) extends conservatively to RoPE by applying it to the position-independent matrix $M = W^Q W^{K\top}$, which empirically upper-bounds all $\|M_{m,n}\|_2$; see Appendix C.4 for details. $\square$

### C.4. Extension of Probabilistic Bounds to RoPE

This section justifies the application of Proposition 3.4 to RoPE architectures.

**The challenge.** Proposition 3.4 bounds overflow probability for a fixed interaction matrix $M = W^Q W^{K\top}$. With RoPE, the effective interaction matrix becomes position-dependent: $M_{m,n} = W^Q R_m^\top R_n W^{K\top}$, where $R_m, R_n$ are position-dependent rotation matrices.

**Why the bound remains conservative.** We apply Proposition 3.4 using $\|M\|_2 = \|W^Q W^{K\top}\|_2$ rather than $\|M_{m,n}\|_2$. This is justified by two observations:

1. **Empirical validation:** We verified that $\max_{m,n} \|M_{m,n}\|_2 \le \|W^Q W^{K\top}\|_2$ holds across all layers in Mistral-7B, Llama-2-13B, and Llama-2-70B. RoPE rotations, which depend only on position indices and fixed frequency bands, do not systematically align with the singular subspaces of $W^Q$ and $W^K$ to amplify the spectral norm.

2. **Conservative application:** By using the unrotated spectral norm $\|W^Q W^{K\top}\|_2$, which empirically upper-bounds all position-dependent variants, our calibration factor $\alpha$ provides at least the claimed overflow probability guarantee.

**Remark on the spherical assumption.** Proposition 3.4 assumes input directions are approximately uniform on

$\mathbb{S}^{d-1}$. This models post-LayerNorm tokens, which have controlled norm and near-isotropic directions in high dimensions. The projection $W^{Q\top}x$ does not preserve uniformity (it concentrates along $W^Q$'s singular vectors), but this makes the actual overflow probability *lower* than our bound predicts: inputs are unlikely to align with both $W^Q$'s and $W^K$'s top singular directions simultaneously. Thus, the spherical assumption yields a conservative bound in practice.

In summary, geometry-aware scaling works identically for RoPE architectures: we compute the spectral norm of the position-independent matrix $W^Q W^{K\top}$ and apply the same calibration procedure.

## D. Spectral Norm Distribution Across Layers

The spectral norm $\sigma_{QK}^{(\ell)} = \|W_{(\ell)}^Q W_{(\ell)}^{K\top}\|_2$ varies substantially across layers within each model, motivating the per-layer scale computation in Eq. (15). Figure 1 shows the layer-by-layer spectral norms for all four architectures, and Table 9 summarizes the distribution statistics.

*Table 9.* Spectral norm statistics across layers for pretrained weights. Max Layer indicates which layer has the largest spectral norm.

| Model | Mean | Max | Min | Max Layer |
|---|---|---|---|---|
| GPT-2 XL | 83.1 | 483.9 | 55.8 | 0 |
| Mistral-7B | 4.9 | 46.8 | 2.4 | 0 |
| Llama-2-13B | 198.4 | 463.5 | 134.4 | 0 |
| Llama-2-70B | 584.2 | 1786.1 | 264.6 | 67 |

The variation is substantial: GPT-2 XL exhibits a $8.7\times$ range (55.8 to 483.9), Mistral-7B a $19.5\times$ range (2.4 to 46.8), Llama-2-13B a $3.5\times$ range (134.4 to 463.5), and Llama-2-70B a $6.7\times$ range (264.6 to 1786.1). This heterogeneity confirms that per-layer scaling is essential: a global scale factor calibrated for the maximum spectral norm would waste FP8 dynamic range on most layers, while one calibrated for the average would risk overflow on high-norm layers.

## E. Algorithm Details

This appendix provides detailed algorithms for the spectral norm estimation procedures described in Section 4.

### E.1. Power Iteration for Standard Multi-Head Attention

Algorithm 2 computes the spectral norm $\|M\|_2$ where $M = W^Q W^{K\top}$ for standard multi-head attention (where $n_q = n_{kv}$). The algorithm maintains persistent vectors $u$ and $v$ that track the top left and right singular vectors across training steps.

---

**Algorithm 2** Power Iteration for Spectral Norm Estimation

1: **Input:** Weight matrices $W^Q, W^K \in \mathbb{R}^{d \times (n_q \cdot d_h)}$
2: **State:** Persistent vectors $u, v \in \mathbb{R}^d$
3: **Output:** Spectral norm estimate $\sigma$
4:
5: **if** not initialized **then**
6:    $u \leftarrow$ random unit vector in $\mathbb{R}^d$
7:    $v \leftarrow$ random unit vector in $\mathbb{R}^d$
8: **end if**
9:
10: {One iteration of power method}
11: $z \leftarrow W^{K\top}v$ {$z \in \mathbb{R}^{n_q \cdot d_h}$}
12: $u' \leftarrow W^Q z$ {$u' = Mv \in \mathbb{R}^d$}
13: $\sigma \leftarrow \|u'\|_2$ {Spectral norm estimate}
14: $u \leftarrow u'/\sigma$ {Update persistent left singular vector}
15:
16: $y \leftarrow W^{Q\top}u$ {$y \in \mathbb{R}^{n_q \cdot d_h}$}
17: $v' \leftarrow W^K y$ {$v' = M^\top u \in \mathbb{R}^d$}
18: $v \leftarrow v'/\|v'\|_2$ {Update persistent right singular vector}
19:
20: **return** $\sigma$

---

**Complexity.** Each iteration requires four matrix-vector products: $W^{K\top}v$, $W^Q z$, $W^{Q\top}u$, and $W^K y$. Each product costs $O(n_q \cdot d_h \cdot d)$ operations, for a total of $O(n_q \cdot d_h \cdot d)$ per iteration. This is far cheaper than forming $M \in \mathbb{R}^{d \times d}$ explicitly, which would cost $O(n_q \cdot d_h \cdot d^2)$.

**Convergence.** Power iteration converges to the top singular value at rate $(\sigma_2/\sigma_1)^k$ after $k$ iterations, where $\sigma_1 \geq \sigma_2$ are the two largest singular values. Since we maintain persistent vectors across forward passes and weights change gradually during training, one iteration per forward pass suffices to track the slowly drifting singular vectors.

### E.2. Implicit GQA Power Iteration

For Grouped Query Attention with $n_q > n_{kv}$, Algorithm 3 computes the spectral norm without expanding the key matrix.

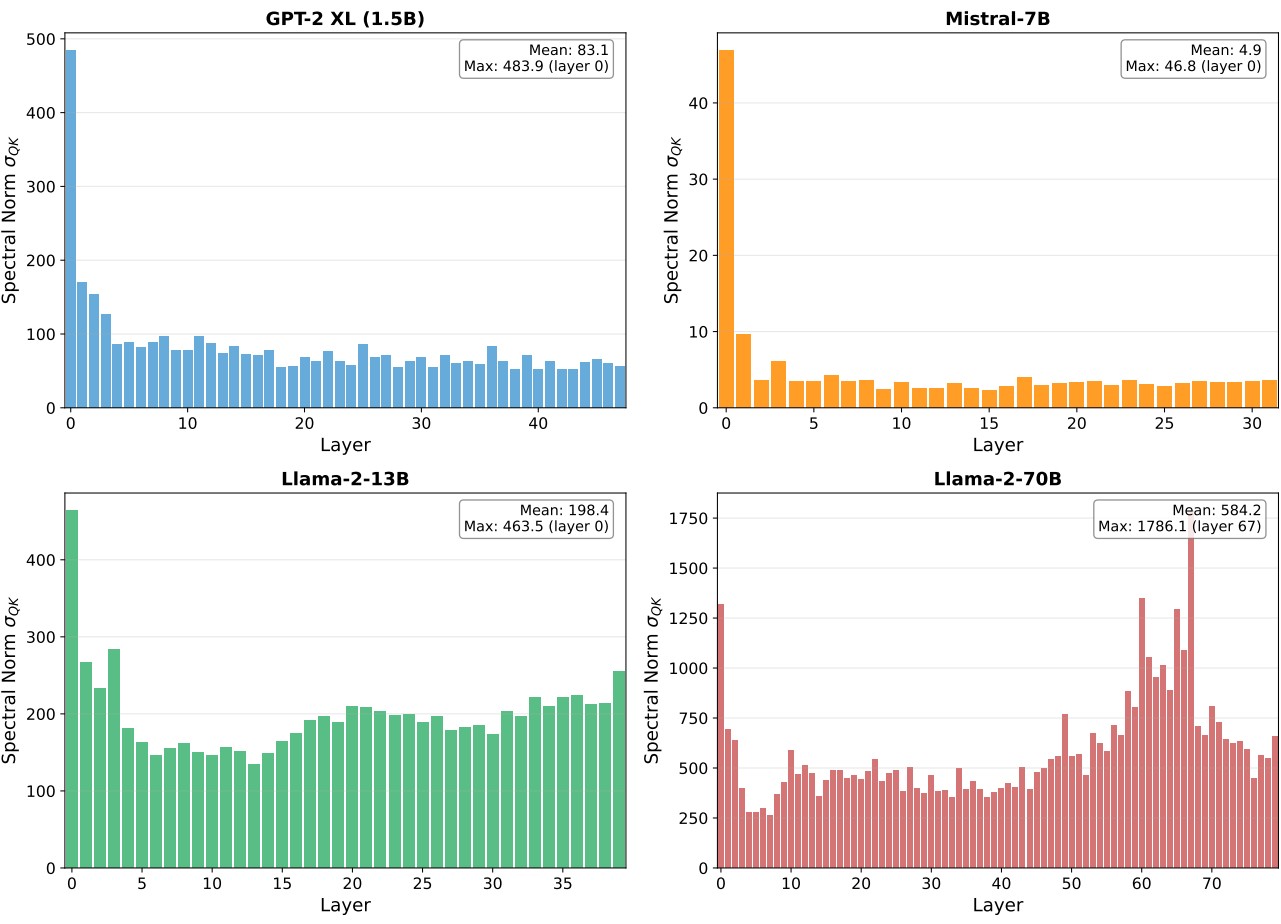

*Figure 1.* Spectral norm $\sigma_{QK}^{(\ell)}$ by layer for all four models, computed on pretrained weights. Early layers consistently exhibit larger spectral norms, with layer 0 being the maximum in three of four models.

---

**Algorithm 3** Implicit GQA Power Iteration

1: **Input:** $W^Q \in \mathbb{R}^{d \times (n_q \cdot d_h)}, W^K \in \mathbb{R}^{d \times (n_{kv} \cdot d_h)}$
2: **Parameters:** Number of query heads $n_q$, KV heads $n_{kv}$, head dimension $d_h$
3: **State:** Persistent vectors $u, v \in \mathbb{R}^d$
4: **Output:** Spectral norm estimate $\sigma \approx \|W^Q W_{\exp}^{K\top}\|_2$
5:
6: $g \leftarrow n_q/n_{kv}$ {Group size}
7:
8: **if** not initialized **then**
9:     $u \leftarrow$ random unit vector in $\mathbb{R}^d$
10:     $v \leftarrow$ random unit vector in $\mathbb{R}^d$
11: **end if**
12:
13: {Forward: compute $Mv$ where $M = W^Q W_{\exp}^{K\top}$}
14: $z_{\text{kv}} \leftarrow W^{K\top} v$ {$z_{\text{kv}} \in \mathbb{R}^{n_{kv} \cdot d_h}$}
15: $z \leftarrow \text{REPEATBLOCKS}(z_{\text{kv}}, g)$ {$z \in \mathbb{R}^{n_q \cdot d_h}$}
16: $u' \leftarrow W^Q z$ {$u' \in \mathbb{R}^d$}
17: $\sigma \leftarrow \|u'\|_2$
18: $u \leftarrow u'/\sigma$
19:
20: {Backward: compute $M^\top u$ where $M^\top = W_{\exp}^K W^{Q\top}$}
21: $y \leftarrow W^{Q\top} u$ {$y \in \mathbb{R}^{n_q \cdot d_h}$}
22: $y_{\text{kv}} \leftarrow \text{SUMGROUPS}(y, g)$ {$y_{\text{kv}} \in \mathbb{R}^{n_{kv} \cdot d_h}$}
23: $v' \leftarrow W^K y_{\text{kv}}$ {$v' \in \mathbb{R}^d$}
24: $v \leftarrow v'/\|v'\|_2$
25:

**Memory savings.** The explicit approach would require expanding $W^K$ from $(n_{kv} \cdot d_h) \times d$ to $(n_q \cdot d_h) \times d$ by replicating columns. For Mistral-7B with $n_q = 32$, $n_{kv} = 8$, $d_h = 128$, and $d = 4096$, this expansion would require $32 \times 128 \times 4096 \times 2 = 32\text{MB}$ per layer (in FP16). The implicit formulation avoids this entirely by operating on the unexpanded $W^K$ and only replicating/summing small intermediate vectors of size $n_{kv} \cdot d_h$.

## F. Proof of Implicit GQA Equivalence

This appendix proves Proposition 4.1, establishing that the implicit GQA formulation computes the same spectral norm as explicit expansion.

*Proof.* Let $g = n_q/n_{kv}$ be the group size. We have $W^Q \in \mathbb{R}^{d \times (n_q \cdot d_h)}$ and $W^K \in \mathbb{R}^{d \times (n_{kv} \cdot d_h)}$. Define $W_{\exp}^K \in \mathbb{R}^{d \times (n_q \cdot d_h)}$ as the matrix obtained by replicating each block of $d_h$ columns of $W^K$ exactly $g$ times. The interaction matrix is $M = W^Q W_{\exp}^{K\top} \in \mathbb{R}^{d \times d}$.

**Forward direction.** For any $v \in \mathbb{R}^d$, we show that $Mv = W^Q \cdot \text{REPEATBLOCKS}(W^{K\top}v, g)$.

Since $W_{\text{exp}}^{K\top} \in \mathbb{R}^{(n_q \cdot d_h) \times d}$ has repeated row-blocks, $W_{\text{exp}}^{K\top}v \in \mathbb{R}^{n_q \cdot d_h}$ consists of $n_q$ blocks of size $d_h$, where blocks $\{(i-1)g+1, \ldots, ig\}$ for $i = 1, \ldots, n_{kv}$ all equal the $i$-th block of $W^{K\top}v \in \mathbb{R}^{n_{kv} \cdot d_h}$. This is exactly the output of $\text{REPEATBLOCKS}(W^{K\top}v, g)$. Since $W^Q \in \mathbb{R}^{d \times (n_q \cdot d_h)}$, multiplying by this repeated vector yields $Mv \in \mathbb{R}^d$.

**Backward direction.** For any $u \in \mathbb{R}^d$, we show that $M^\top u = W^K \cdot \text{SUMGROUPS}(W^{Q\top}u, g)$.

We have $M^\top = W_{\text{exp}}^K W^{Q\top}$. First, $W^{Q\top}u \in \mathbb{R}^{n_q \cdot d_h}$. Since $W_{\text{exp}}^K \in \mathbb{R}^{d \times (n_q \cdot d_h)}$ has $g$ identical copies of each column block of $W^K$, multiplying $W_{\text{exp}}^K$ by a vector $y \in \mathbb{R}^{n_q \cdot d_h}$ is equivalent to multiplying $W^K$ by the vector that sums each group of $g$ blocks: $W_{\text{exp}}^K y = W^K \cdot \text{SUMGROUPS}(y, g)$. Thus $M^\top u = W^K \cdot \text{SUMGROUPS}(W^{Q\top}u, g) \in \mathbb{R}^d$.

**Conclusion.** Power iteration maintains $u, v \in \mathbb{R}^d$ and computes $Mv$ and $M^\top u$ via the implicit formulation. Since these equal the explicit products, the iteration converges to the same singular vectors and value: $\sigma_{\text{implicit}} = \sigma_{\text{explicit}} = \|W^Q W_{\text{exp}}^{K\top}\|_2$. $\square$

## G. Experimental Details

This appendix provides additional experimental details and extended results for Section 5.

### G.1. Training Configuration

Table 10 summarizes the architectural specifications for all four models evaluated.

*Table 10.* Model architectures. MHA: multi-head attention. GQA: grouped query attention.

|  | **GPT-2 XL** | **Mistral-7B** | **Llama-2-13B** | **Llama-2-70B** |
|---|---|---|---|---|
| Params | 1.5B | 7B | 13B | 70B |
| Layers | 48 | 32 | 40 | 80 |
| Attention | MHA | GQA | MHA | GQA |
| $n_q$ / $n_{kv}$ | 25/25 | 32/8 | 40/40 | 64/8 |
| $d$ | 1600 | 4096 | 5120 | 8192 |
| $d_h$ | 64 | 128 | 128 | 128 |
| Hardware | H100 | H200 | H200 | B200 |

Table 11 provides the full training configuration for each model.

**Delayed scaling baseline configuration.** For the delayed scaling baseline, we use the standard configuration from Micikevicius et al. (2022): history buffer length of 16 steps and safety margin $\eta = 0.9$ (see Eq. 1). The history buffer is initialized to 1.0 at the start of training or after checkpoint

*Table 11.* Training configuration for all experiments.

|  | **GPT-2 XL** | **Mistral-7B** | **Llama-2-13B** | **Llama-2-70B** |
|---|---|---|---|---|
| Batch size | 8 | 4 | 4 | 2 |
| Seq. length | 1024 | 1024 | 1024 | 1024 |
| Learning rate | $10^{-4}$ | $10^{-5}$ | $10^{-5}$ | $10^{-5}$ |
| Optimizer | AdamW | AdamW | AdamW | AdamW |
| Weight decay | 0.01 | 0.01 | 0.01 | 0.01 |
| Grad. clip | 1.0 | 1.0 | 1.0 | 1.0 |
| FP8 format | E4M3 | E4M3 | E4M3 | E4M3 |
| $\alpha$ | 0.08 | 0.04 | 0.03 | 0.02 |
| $\eta_{\text{fp8}}$ | 0.8 | 0.8 | 0.8 | 0.8 |

loading, which is the source of history staleness in transient scenarios.

## H. Instantaneous Response Validation

This appendix validates the instantaneous response property of geometry-aware scaling with a controlled stress test.

**Setup.** We train GPT-2 XL normally for 10 steps, then multiply all attention weights by $4\times$ at step 10, simulating an extreme transient. This artificial perturbation isolates the mechanism: delayed scaling relies on stale statistics and cannot adapt, while geometry-aware scaling recomputes the bound from current weights.

**Results.** Figure 2 shows the results. Delayed scaling, relying on statistics that predate the weight change, produces scaled logits exceeding 6500 and overflows catastrophically. Geometry-aware scaling responds in the same forward pass: the scale factor jumps from 5.8 to 93.8 at step 10, keeping scaled logits below 85 throughout.

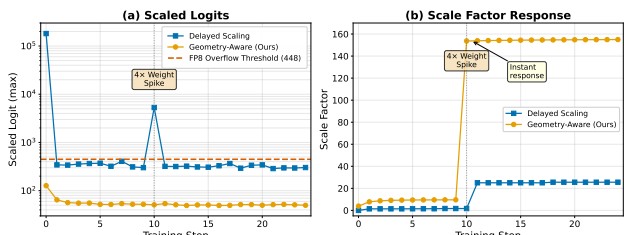

*Figure 2.* Response to $4\times$ weight spike at step 10 (GPT-2 XL). (a) Maximum scaled attention logit. (b) Scale factor over time. Delayed scaling overflows because its history buffer contains no information about the weight change. Geometry-aware scaling adapts instantaneously because the scale factor is computed from current weights.

While the $4\times$ spike is a stress test, the underlying mechanism explains why geometry-aware scaling succeeds in the realistic transient scenarios evaluated in Section 5.2: loading pretrained checkpoints, checkpoint resumption, and learning rate transitions all involve weight configurations for which the history buffer has no relevant statistics.

## I. Overhead Measurement

Table 12 reports forward pass timing for all models.

*Table 12.* Computational overhead (forward pass time, averaged over 100 iterations).

| Model | Attention | Delayed | Ours | Overhead |
|---|---|---|---|---|
| GPT-2 XL | MHA | 136.3 ms | 137.6 ms | +1.0% |
| Mistral-7B | GQA 4:1 | 58.2 ms | 55.1 ms | −**5.3**% |
| Llama-2-13B | MHA | 96.2 ms | 98.1 ms | +1.9% |
| Llama-2-70B | GQA 8:1 | 206.3 ms | 215.0 ms | +4.3% |

Measurements follow this protocol:

1. Warm up GPU with 10 forward passes (discarded)

2. Run 100 forward passes with CUDA synchronization after each

3. Report mean time per forward pass

4. Repeat 3 times and report median across repetitions

For fair comparison, both methods use identical model configurations, batch sizes, and sequence lengths. The only difference is the scale factor computation method.

## J. FP8 Utilization Statistics

Table 13 shows FP8 dynamic range utilization during training with different methods on GPT-2 XL.

*Table 13.* FP8 dynamic range utilization (%) during training (GPT-2 XL).

| Method | Median | P10 | P90 |
|---|---|---|---|
| Delayed | 90.5% | 84.0% | 98.0% |
| Ours (conservative) | 14.3% | 13.6% | 15.3% |
| Ours + auto-$\alpha$ | 31.2% | 28.5% | 33.7% |

## K. MMLU Training Curves

Figure 3 shows training loss curves for the three scaling methods on Llama-2-13B fine-tuned on MMLU STEM subjects.

The key observation is that training loss alone does not predict downstream performance: the conservative variant achieves comparable loss (0.0113 vs. 0.0112) but about 4 percentage points lower MMLU accuracy (28.7% vs. 32.6%). This gap arises because excessive quantization noise from 0.5% FP8 utilization degrades the fine-grained attention patterns required for reasoning tasks, even when average loss remains low.

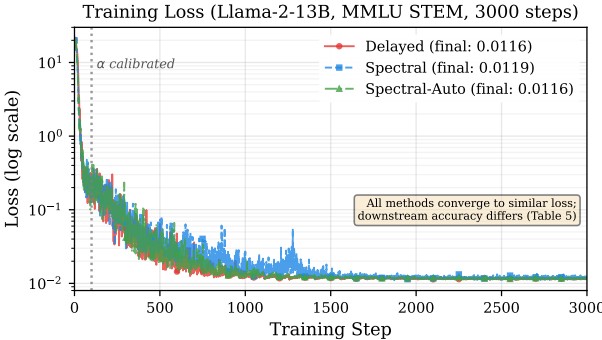

*Figure 3.* Training loss comparison for delayed scaling, geometry-aware scaling (conservative), and geometry-aware scaling with auto-$\alpha$ on Llama-2-13B (MMLU STEM, 10000 steps). All methods converge to similar final loss ($\approx 0.0112$), yet downstream MMLU accuracy differs substantially (Table 5). The vertical dashed line indicates when auto-$\alpha$ calibration completes (step 100). The conservative variant shows slightly higher loss throughout training due to reduced FP8 utilization (0.5% vs. 34.9% for auto-$\alpha$).

## L. Detailed MMLU Results by Subject

Table 14 shows accuracy on all 17 MMLU STEM subjects.

*Table 14.* MMLU accuracy (%) by subject on Llama-2-13B after 10000 fine-tuning steps.

| Subject | Delayed | Cons. | Auto-$\alpha$ |
|---|---|---|---|
| Abstract Algebra | 26.0 | 30.0 | 18.0 |
| College Math | 33.0 | 25.0 | 33.0 |
| Elementary Math | 25.4 | 26.2 | 27.5 |
| HS Math | 23.0 | 21.5 | 21.5 |
| HS Statistics | 27.8 | 27.8 | 31.5 |
| Astronomy | 29.6 | 36.2 | 42.1 |
| College Physics | 39.2 | 38.2 | 41.2 |
| HS Physics | 27.8 | 26.5 | 27.2 |
| College CS | 36.0 | 32.0 | 34.0 |
| Computer Security | 41.0 | 35.0 | 37.0 |
| HS CS | 37.0 | 30.0 | 41.0 |
| College Chemistry | 29.0 | 34.0 | 33.0 |
| HS Chemistry | 34.0 | 28.6 | 30.5 |
| College Biology | 46.5 | 27.8 | 39.6 |
| HS Biology | 45.5 | 32.6 | 43.9 |
| Electrical Eng. | 34.5 | 30.3 | 33.8 |
| Machine Learning | 28.6 | 17.0 | 28.6 |
| **Average** | **32.6** | **28.7** | **32.7** |

The largest gaps in favor of auto-$\alpha$ over conservative

spectral appear in College Biology (+11.8 pp), Machine Learning (+11.6 pp), HS Biology (+11.3 pp), and HS Computer Science (+11.0 pp). This pattern is consistent with excessive quantization noise affecting tasks that rely on fine-grained attention patterns. Per-subject test sets are small (some as few as 9–16 examples), so individual deltas are noisy, but the direction is consistent: auto-$\alpha$ matches delayed scaling on average and recovers utilization without sacrificing accuracy.

## M. Auto-$\alpha$ Calibration Details

### M.1. Algorithm

Algorithm 4 provides the complete auto-$\alpha$ calibration procedure.

---

**Algorithm 4** Auto-$\alpha$ Calibration

---

**Require:** Conservative initial $\alpha_0$, burn-in steps $T_{\text{calib}}$, quantile $q$, safety multiplier $\kappa$
1: Initialize slack ratio buffer $R \leftarrow \emptyset$
2: **for** $t = 1$ to $T_{\text{calib}}$ **do**
3:     Compute $B_{\max}^{(t)} = \|W^Q W^{K\top}\|_2 \cdot d/\sqrt{d_h}$
4:     Run forward pass with scale = $\alpha_0 \cdot B_{\max}^{(t)}/(0.8 \times 448)$
5:     Record $M_t = \max_{i,j} |S_{ij}^{(t)}|$ (actual maximum logit)
6:     Append $r_t = M_t/B_{\max}^{(t)}$ to $R$
7: **end for**
8: $\alpha_{\text{emp}} \leftarrow \text{Quantile}_q(R)$
9: $\alpha_{\text{final}} \leftarrow \alpha_{\text{emp}} \times \kappa$
10: **Freeze** $\alpha_{\text{final}}$ for all subsequent training

---

### M.2. Calibration Statistics

During the 100-step burn-in on Llama-2-13B:

- Slack ratio range: $[7.3 \times 10^{-5}, 4.7 \times 10^{-4}]$
- Slack ratio mean: $1.9 \times 10^{-4}$
- $P_{99.99}$ quantile: $4.6 \times 10^{-4}$
- Calibrated $\alpha$: $4.6 \times 10^{-4}$ (with $\kappa = 1.0$)

This represents a 65× tightening compared to the conservative $\alpha = 0.03$, resulting in proportionally higher FP8 utilization (34.9% vs. 0.5%).

### M.3. Safety Multiplier Selection

The safety multiplier $\kappa$ controls the trade-off between utilization and robustness:

- $\kappa = 1.0$: Maximum utilization, suitable for stable fine-tuning

- $\kappa = 2.0$: Moderate headroom for distribution shift

- $\kappa = 3.0$: Conservative, suitable for heterogeneous data

In our experiments, $\kappa = 1.0$ achieved zero overflows across 10000 steps, suggesting the $P_{99.99}$ quantile provides sufficient margin for MMLU STEM fine-tuning.

