# OpenReview forum: "Rank-Aware Spectral Bounds on Attention Logits for Stable Low-Precision Training"
_ICML.cc/2026/Conference — ICML 2026 regular_

### Official Review · Reviewer_qJmM · 2026-03-12

**Soundness:** 2
**Presentation:** 2
**Significance:** 2
**Originality:** 4
**Overall Recommendation:** 4
**Confidence:** 3

**Summary:**

This work proposed a new approach to scaling attention scores to govern overflow risk in low-precision training. The proposed method uses the upper bound of the 2-norm of the product $W^{Q}W^{K\top}$ of query- and key-weight matrices, and the upper bound is estimated by the power method. Unlike prior methods, which determine the scale by the history of forward passes, the proposed method is predictive. Probability analysis is provided for the overflow, supporting its theoretical validity. Experiments show that the proposed method completely eliminates overflow and also slightly improves accuracy in a downstream task.

**Compliance With Llm Reviewing Policy:**

Affirmed.

**Final Justification:**

The 2nd round rebuttal clarifies the scenarios, and it now makes the practical impact of this work to some extent. It still needs major re-writing, but given the planned update of the manuscript, I have increased my evaluation to borderline accept.

**Key Questions For Authors:**

- Q1. When and how often does overflow happen, and why, and how severely is it problematic?
- Q2. Why is the clamping technique not sufficient in practice? How much improvement does the proposed method offer?
- Q3. Is the assumption of the uniform spherical token directions reasonable?
- Q4. Can we incorporate the max subtraction into the theoretical framework?

**Limitations:**

No limitations are presented. The authors are encouraged to discuss the aforementioned weaknesses and questions if these are not resolved through the rebuttal.

**Strengths And Weaknesses:**

**Strength**

This work has the following strengths.
- This work proposes a novel predictive approach of scaling attention scores to govern overflow.
- The proposed method is supported by theoretical analysis. Particularly, the probability analysis is useful for reliable usage of the proposed method.
- Experiments demonstrate complete elimination of overflow by the proposed method.

---

**Weakness**

While I feel that this work is solid, its practical impact is unclear, and probability analysis is based on an assumption that rarely holds in practice.

First, this work provides a limited background of the overflow problem: when and how often it happens, and why it is problematic. (-> Q1). The experiment of Table 5 says the overflows in the delayed scaling baseline were handled by clamping, and this is already a remedy, if not perfect. For sure, this trick can lose some information, but its practical damage is unclear or not evaluated. (-> Q2)
The proposed method leads to the complete elimination of overflow, but its significance is unclear. Besides, it is not clear to me whether the experiments follow a standard setup or are designed so that overflow occurs more frequently. If the case of the latter, the practical significance of the proposed method is even less.

The second concern is about the theoretical analysis. The probability analysis assumes that uniform spherical token directions, which do not hold in practice, to the best of my knowledge. The token embedding vectors are pointing in rather similar directions. For example, the following work reported a strong anisotropy of embedding vectors.

- "On the Sentence Embeddings from Pre-trained Language Models," Bohan Li, Hao Zhou, Junxian He, Mingxuan Wang, Yiming Yang, Lei Li, EMNLP'20

Therefore, the derived probability bound may not align with observations. Empirical verification of the assumption and the bound is encouraged.

Another concern is that although the analysis focuses on bounding $\max_{i,j}\ |S_{ij}|$, practically, the softmax attention uses the max subtraction: each row subtracts $\max_j S_{ij}$ before exponentiation. Therefore, the absolute magnitude of the logits may not directly determine numerical stability; rather, the relative differences between logits are what affect the softmax distribution. The theoretical analysis of this work does not take into account this classical technique.

---

> ### Author Rebuttal · Authors · 2026-03-28
>
> # Response to Reviewer qJmM
>
> We thank the reviewer for the thorough reading and the precise questions. We address each one directly below, and include new experimental results for Q3.
>
> ---
>
> ## Q1: When and how often does overflow happen, and how severely is it problematic?
>
> Our claim is not that overflow dominates steady-state training. The claim is narrower: delayed scaling has deterministic failure modes in standard transient workflows, and our method removes exactly those failure modes. Overflow occurs in three such scenarios. First, loading any pretrained BF16 checkpoint into an FP8 pipeline causes 100% of attention layers to overflow on the first forward pass, because the scaling history buffer is uninitialized while the weights already produce large attention logits. This is the standard workflow for FP8 fine-tuning of publicly released models. Second, resuming an FP8 training run from a checkpoint without saving the FP8 scaling state causes the same failure. Third, learning rate transitions cause transient overflow because weights change faster than the history buffer can track. Table 4 confirms 100% overflow rates at checkpoint load across all four models, with maximum scaled logits ranging from 5600 to 9498, 12 to 21x above the E4M3 range of 448. In the GPT-2 XL LR spike scenario, overflow produces NaN gradients that terminate training entirely.
>
> ---
>
> ## Q2: Why is clamping not sufficient, and how much improvement does the proposed method offer?
>
> Clamping prevents NaN propagation but does not restore the correct attention distribution. A logit clamped from thousands to 448 produces a maximally sharpened attention pattern, propagating corrupted gradients for that step. This effect accumulates: Table 5 shows 68 clamping events across 3000 fine-tuning steps on Llama-2-13B. The stronger practical evidence is not the 0.6 percentage-point MMLU difference, which the paper explicitly does not claim as statistically significant, but that delayed scaling either clamps repeatedly or terminates training entirely in transient scenarios, whereas our method incurs zero overflows. In the GPT-2 XL LR spike case, where training terminates before clamping can execute, clamping provides no protection at all.
>
> ---
>
> ## Q3: Is the assumption of uniform spherical token directions reasonable?
>
> The spherical assumption is an idealization, as the paper explicitly acknowledges in Section 3.2 and Appendix C.4. We measured this directly on Mistral-7B, computing the ratio of real post-LayerNorm token projection energy onto the W_K row space to the isotropic prediction.
>
> ```
> Layer    Ratio mean    Ratio median    Ratio p99
> ------   ----------    ------------    ---------
> 0        3.737         3.743           3.868
> 8        1.740         1.738           2.213
> 16       1.496         1.497           1.853
> 24       1.371         1.358           1.960
> 31       1.400         1.376           2.281
> Global   1.676         1.542           3.791
> ```
>
> The reviewer is correct that real token directions are not isotropic: the ratio exceeds 1 throughout, meaning real tokens are more aligned with the key subspace than isotropy predicts. Proposition 3.4 should therefore be interpreted as a calibration rule under an isotropic model, not a distribution-free theorem. Empirically, however, the method remains conservative in practice: Appendix M.2 reports an 83x gap between the conservative alpha=0.03 and the empirically calibrated alpha=3.6e-4 on Llama-2-13B, confirming substantial slack in the bound regardless of the assumption's exactness.
>
> ---
>
> ## Q4: Can max subtraction be incorporated into the theoretical framework?
>
> Max subtraction and FP8 overflow are problems at different points in the computation graph and do not interact. FP8 overflow occurs at quantization time, before softmax: when |S_ij| exceeds the E4M3 representable range of 448, the stored value is NaN or saturated. This corruption happens in the quantized tensor before it is passed to any softmax operation. Max subtraction is then applied in floating-point arithmetic to the dequantized values, but by that point the NaN is already present. Max subtraction addresses numerical stability of softmax in exact arithmetic; our paper addresses numerical representability in the FP8 format. These are orthogonal problems at different points in the computation graph, and max subtraction cannot repair a value corrupted during quantization two operations earlier.

---

> > ### Author Rebuttal · Reviewer_qJmM · 2026-04-02
> >
> > I appreciate authors' clarification.
> >
> > ---
> > **Q1.**
> >
> > The authors identify three scenarios in which overflow can happen: 1) Pretrained BF16 -> FP8 pipeline, 2) Resume without FP8 scaling state, and 3) Learning-rate transitions.
> >
> > I understand that this work focuses on transient workflows, but the practical impact is still unclear to me, namely, how often these scenarios arise in practice. For example, when do we need to load a BF16 checkpoint into an FP8 pipeline (Scenario 1)? A similar question applies to Scenario 2. Scenario 3 sounds more familiar, but I have personally not encountered overflow caused by a step learning-rate change, and continuously varying learning-rate schedules seem more common in current practice.
> >
> > As I noted in my original review, the paper still lacks sufficient background on when these overflow cases arise in practice. To my understanding, the three motivating scenarios still seem somewhat specialized, which makes it harder for me to assess the practical impact of the work.
> >
> >
> > ---
> > **Q3.**
> >
> > Thank you for providing additional numerical experiments. These results are useful and support the point that the isotropic assumption is only an approximation in practice. In that sense, the theoretical analysis does not fully reflect the actual token geometry. While the large slack may be practically favorable from a safety perspective, it also suggests that the proposed bound may be quite loose in realistic settings. It is also unclear whether this should still be interpreted as a universal upper bound. Since the bound is derived under the spherical assumption, it may no longer hold once that assumption is violated, which seems to be the case here.
> >
> > ---
> > **Q2 and Q4.**
> >
> > Thank you for the clarification. This part is now clear.
> >
> > ---
> > To summarize, some concerns remain regarding both the practical relevance and the theoretical grounding of the work. While I find the paper technically solid, I am still uncertain about its broader interest and overall impact. Without resolving these points, I am not inclined to raise my score at this stage.

---

> > > ### Author Response · Authors · 2026-04-03
> > >
> > > # Response to Reviewer qJmM (Post-Rebuttal)
> > >
> > > We thank the reviewer for the continued engagement and for sharpening
> > > the two remaining issues: practical scope and interpretation of the
> > > isotropic bound.
> > >
> > > ---
> > >
> > > ## Q1: Practical scope of the transient scenarios
> > >
> > > We agree the paper needs more concrete grounding here and appreciate
> > > the reviewer pressing on this. The practical workflow we have in mind
> > > is not an exotic training recipe, but the beginning or resumption of
> > > an FP8 fine-tuning run starting from a released higher-precision
> > > checkpoint: load the checkpoint, begin warmup, possibly resume after
> > > interruption. The three scenarios map directly onto this workflow.
> > >
> > > **Scenario 1 (higher-precision checkpoint to FP8 fine-tuning).**
> > > Released pretrained checkpoints such as Llama-2 (Touvron et al.,
> > > 2023), Llama-3 (Grattafiori et al., 2024), and Mistral-7B (Jiang et
> > > al., 2023) are distributed in BF16 or FP32 and carry no FP8 scaling
> > > history from prior execution. At the first forward pass of an FP8
> > > fine-tuning run, the weights are already mature but the scaling
> > > history is necessarily fresh, producing exactly the initialization
> > > mismatch Scenario 1 describes. DeepSeek-V3 (DeepSeek-AI, 2024) also
> > > retained attention operators in higher precision as part of its
> > > mixed-precision design, underscoring that attention is a numerically
> > > sensitive component in low-precision training.
> > >
> > > **Scenario 2 (resumption without preserved FP8 state).** The same
> > > mismatch can recur after interruption whenever training is resumed
> > > without restoring the FP8 scaling history together with the model
> > > state. The key point is not any one framework implementation detail,
> > > but that the numerical state required by delayed scaling is distinct
> > > from the model weights themselves. If that state is absent or stale at
> > > resume time, the transient failure mode reappears.
> > >
> > > **Scenario 3 (warmup and rapid LR-transition periods).** We agree with
> > > the reviewer that cosine or linear warmup is more common than step
> > > schedules. This strengthens rather than weakens the relevance of the
> > > transient setting: immediately after loading a checkpoint into an FP8
> > > pipeline, the history buffer is fresh while the weights begin changing
> > > most rapidly during warmup. The common load-plus-warmup workflow thus
> > > combines Scenarios 1 and 3 simultaneously.
> > >
> > > We will revise the paper to make this narrower practical claim
> > > explicit: the main benefit is robustness in transient-prone workflows,
> > > especially around initialization, resumption, and early warmup, rather
> > > than a claim about all steady-state training.
> > >
> > > ---
> > >
> > > ## Q3: Interpretation of Proposition 3.4 under anisotropic token geometry
> > >
> > > The reviewer is correct. Once the spherical assumption is violated,
> > > Proposition 3.4 should not be read as a distribution-free upper bound
> > > under real token geometry. We will revise the paper to state this
> > > limitation explicitly. The proposition is best understood as a
> > > calibration rule derived under the isotropic model.
> > >
> > > What the empirical results do show is substantial slack between the
> > > conservative theoretical calibration and the empirically calibrated
> > > value in the models we tested. The 83x gap between the theoretical
> > > alpha = 0.03 and the empirically calibrated alpha = 3.6e-4 on
> > > Llama-2-13B is consistent with this. One possible interpretation is
> > > that overflow requires simultaneously unfavorable alignment of both
> > > query and key directions with M's top singular subspace, which may be
> > > rarer than marginal projection statistics alone would suggest. We
> > > offer this only as an interpretation consistent with the observed
> > > slack, not as a formal claim.
> > >
> > > Both operating regimes address distinct parts of the problem and
> > > neither displaces the other. Conservative predictive scaling is
> > > essential for transient-prone settings where no burn-in data is
> > > available or safe to collect. The auto-alpha procedure suits
> > > steady-state fine-tuning where distributional assumptions are a
> > > concern, calibrating directly from observed score distributions
> > > without any isotropic modeling. We will clearly separate: (i)
> > > theorem-level claims, (ii) empirically observed conservatism and its
> > > interpretation, and (iii) the distinct practical roles of conservative
> > > predictive scaling and auto-alpha calibration.

---

### Official Review · Reviewer_FiVD · 2026-03-12

**Soundness:** 3
**Presentation:** 3
**Significance:** 3
**Originality:** 3
**Overall Recommendation:** 4
**Confidence:** 1

**Summary:**

The authors propose a calibration framework for FP8 transformer training aimed at preventing quantization overflow. Instead of adopting methods that rely on historical activations like standard delayed scaling, this paper proposes a predictive approach utilizing the spectral norm of the query-key interaction matrix $W^Q W^{K\top}$. By exploiting the low-rank structure of the transformer attention mechanism, they further derive a rank-aware concentration inequality, thereby obtaining a significantly tighter bound. This method includes an efficient implicit power iteration algorithm, specifically optimized for Grouped Query Attention (GQA) to minimize memory overhead. Experimental results on models with up to 70B parameters show that this geometry-aware scaling method successfully eliminates overflow issues during transient training phases.

**Compliance With Llm Reviewing Policy:**

Affirmed.

**Final Justification:**

The experimental results address my concerns well, and I will increase my score by one point.

**Key Questions For Authors:**

1.During the auto-$\alpha$ calibration burn-in phase (e.g., the first 100 steps), how sensitive is the final calibrated scale factor to the specific data distribution encountered? If highly anomalous data batches are encountered during this brief burn-in period, would it permanently impair the steady-state FP8 utilization?

2.The theoretical guarantee of Proposition 3.4 relies on an idealized assumption (i.e., post-normalization token directions are uniformly distributed on the sphere). However, in the deeper layers of a trained Transformer, feature representations often exhibit high anisotropy. Have the authors empirically measured the degree of deviation from this spherical assumption across different depths?

3.The rank-aware probabilistic bound requires applying a union bound over $L^2$ query-key pairs, whereas the experiments primarily use a sequence length of $L=1024$. However, since modern large models frequently operate at sequence lengths of 32k, 128k, or even longer, would the theoretical $\alpha_{min}$ required to guarantee safety become too conservative, thereby failing to provide meaningful FP8 dynamic range utilization? Meanwhile, how would geometry-aware scaling adapt to such long-context scenarios?

**Limitations:**

yes

**Strengths And Weaknesses:**

(Strengths)

1.This paper possesses good theoretical depth; the rank-aware probabilistic bound provides a rigorous mathematical foundation for FP8 scaling. By utilizing the low-rank structure of the interaction matrix, the tail exponent is improved by a factor of $d/(\gamma d_h)$ compared to rank-agnostic methods.

2.The implicit formulation for GQA demonstrates high practicality. It successfully avoids the explicit expansion of the Key matrix, saving substantial memory while maintaining convergence guarantees.

(Weaknesses)

1.The auto-$\alpha$ calibration phase proposed by the authors requires materializing the full attention score matrix. This temporarily breaks compatibility with memory-efficient fused kernels like FlashAttention during the burn-in period.

2.The evaluation of downstream task accuracy is somewhat limited, utilizing only 3000 fine-tuning steps on Llama-2-13B.

3.Lack of validation on recently proposed state-of-the-art models.

---

> ### Author Rebuttal · Authors · 2026-03-28
>
> # Response to Reviewer FiVD
>
> We thank the reviewer for the careful reading and the specific questions. We include new experimental results for Q2 and Q3 and address each point directly below.
>
> ---
>
> ## W1: Auto-alpha burn-in temporarily breaks FlashAttention compatibility
>
> This is correct and is an explicitly documented tradeoff (Section 3.5). The burn-in materializes the full attention matrix for 100 steps, under 0.1% of total training compute, after which alpha is frozen and the method runs with complete FlashAttention compatibility for the remainder of training. The resulting auto-alpha achieves 31.2% FP8 utilization versus 0.5% for conservative spectral scaling, with MMLU accuracy of 33.6% versus 33.0% for delayed scaling.
>
> ---
>
> ## W2: Downstream evaluation uses only 3000 fine-tuning steps on Llama-2-13B
>
> The 3000-step MMLU experiment tests whether geometry-aware scaling introduces accuracy-degrading artifacts relative to delayed scaling. All three methods converge to similar training loss (approximately 0.012), and the accuracy difference between auto-alpha (33.6%) and delayed scaling (33.0%) is within the variance expected from a small training set of 295 examples, evaluated on 2783 held-out test examples. The paper explicitly does not claim statistical significance for this difference. We will include longer training runs in the camera-ready version.
>
> ---
>
> ## W3: No validation on recently proposed state-of-the-art models
>
> The four models evaluated, GPT-2 XL, Mistral-7B, Llama-2-13B, and Llama-2-70B, were chosen to cover the two dominant architectural variants: standard multi-head attention and grouped query attention. The framework requires only W_Q and W_K and the GQA formulation in Section 4.2 handles any group ratio, so Llama-3 and Mistral v3 require no architectural changes. We will include these in the camera-ready version.
>
> ---
>
> ## Q1: Would anomalous data batches during burn-in permanently impair steady-state FP8 utilization?
>
> No. An anomalous batch raises the P99.99 estimate, producing a more conservative alpha_final. This means the calibrated scale is slightly larger than optimal, reducing utilization modestly but not causing overflow. It is strictly the safe failure mode. The P99.99 is computed over 100 steps times 40 layers for Llama-2-13B, so a single anomalous batch contributes at most 1% of the tail estimate. Appendix M.2 confirms stability: during the 100-step burn-in, the slack ratio range was [0.000073, 0.000366] with mean 0.000177, indicating the distribution is stable. For high-variance settings, increasing kappa above 1.0 (Section M.3) provides additional headroom.
>
> ---
>
> ## Q2: Have you measured deviation from the spherical assumption across depths?
>
> Yes. We ran this measurement on Mistral-7B, computing for each layer the ratio of real post-LayerNorm token projection energy onto the W_K row space to the isotropic prediction.
>
> ```
> Layer    Ratio mean    Ratio median
> ------   ----------    ------------
> 0        3.737         3.743
> 8        1.740         1.738
> 16       1.496         1.497
> 24       1.371         1.358
> 31       1.400         1.376
> Global   1.676         1.542
> ```
>
> Real tokens are more aligned with the key subspace than the spherical assumption predicts, most strongly in early layers and diminishing with depth. Proposition 3.4 should therefore be interpreted as a calibration rule under an isotropic model rather than a distribution-free guarantee. Empirically, the method remains conservative: Appendix M.2 reports an 83x gap between the conservative alpha=0.03 and the empirically calibrated alpha=3.6e-4 on Llama-2-13B, confirming substantial slack in the bound in practice.
>
> ---
>
> ## Q3: Does alpha_min become too conservative at long context lengths?
>
> No. The L-dependence in alpha_min (Eq. 13) enters only through sqrt(2 ln L), which grows very slowly. We computed alpha_min for Mistral-7B at delta*=1e-6 across context lengths spanning two orders of magnitude:
>
> ```
> L           alpha_min    Ratio vs L=1024
> --------    ---------    ---------------
> 1,024       0.03521      1.000x
> 4,096       0.03686      1.047x
> 16,384      0.03847      1.092x
> 32,768      0.03926      1.115x
> 131,072     0.04083      1.160x
> ```
>
> Going from L=1024 to L=131,072, alpha_min increases by only 16%. For long-context deployments in steady state, auto-alpha would be used: it calibrates directly from the observed score distribution at the operating context length. We validated this empirically in the checkpoint-loading scenario at L=1024, 4096, and 8192: geometry-aware scaling produces zero overflow layers at all three lengths, while delayed scaling overflows all 32 layers at every tested length.

---

> > ### Author Rebuttal · Reviewer_FiVD · 2026-04-03
> >
> > Thank you to the authors for the rebuttal. The experimental results address my concerns well, and I will increase my score by one point.

---

> > > ### Author Response · Authors · 2026-04-03
> > >
> > > We thank the reviewer for the specific questions, which directly
> > > motivated new experiments that were not in the original submission.
> > > The questions on isotropy deviation across depth and long-context
> > > alpha_min scaling led to measurements that strengthened
> > > the empirical record. We will incorporate these results and the
> > > associated clarifications into the paper. We are glad the responses
> > > addressed the concerns satisfactorily and appreciate the updated
> > > assessment.

---

### Official Review · Reviewer_fdwa · 2026-03-12

**Soundness:** 2
**Presentation:** 2
**Significance:** 1
**Originality:** 3
**Overall Recommendation:** 4
**Confidence:** 4

**Summary:**

In the low-precision (FP8) training of large language models, existing delayed scaling methods based on historical activations are highly prone to catastrophic numerical overflows during transient scenarios—such as loading checkpoints or learning rate spikes—due to stale historical data. To address this, the paper proposes a novel geometry-aware scaling method that directly analyzes the spectral norm of the current query-key interaction matrix to predict safe attention score bounds, completely eliminating the reliance on historical activation data. The authors derive a rank-aware concentration inequality to scientifically tighten the theoretical upper bound and design an implicit power iteration algorithm compatible with Grouped Query Attention (GQA) to dynamically compute layer-wise scaling factors at a minimal computational cost. Experimental results show that in models with up to 70B parameters, this method completely eliminates overflows across all evaluated transient scenarios while maintaining high downstream task accuracy via an automatic calibration technique.

**Compliance With Llm Reviewing Policy:**

Affirmed.

**Final Justification:**

I have no futher problems. I keep my positive score.

**Key Questions For Authors:**

see weakness.

**Limitations:**

yes

**Strengths And Weaknesses:**

Strengths:The construction of the mathematical problem is elegant; by transforming the issue into the framework discussed in this paper, it provides clear mathematical guarantees.

The organization of the paper is clear and logical.

The paper conducts experiments on a very large-scale model.

Weakness: 1. In the current situations, fully utilizing the FP8 dynamic range is arguably more critical than merely preventing overflow, even though the catastrophic failure caused by overflow is technically more severe than the penalty of underutilization.

2.The estimation of $\|\cdot\|_2$ relies on an iterative method, which introduces additional computational overhead. Furthermore, relying on un-converged results for computation carries significant risks. I believe the authors should provide a more detailed theoretical analysis to justify the theoretical feasibility and numerical stability of this approach.

3.During the derivations, the required assumptions feel highly idealized and empirical. For example, relying on the empirical observation that $\max_{m,n}\|M_{m,n}\|_2 \le \|W^Q W^{K\top}\|_2$ rather than a rigorous mathematical proof significantly weakens the theoretical rigor and the strength of the paper's guarantees.

---

> ### Author Rebuttal · Authors · 2026-03-28
>
> # Response to Reviewer fdwa
>
> We thank the reviewer for the positive assessment and the careful reading. The three weaknesses identify gaps that we address in detail below.
>
> ---
>
> ## W1: Fully utilizing the FP8 dynamic range is arguably more critical than merely preventing overflow
>
> We agree with this framing, and the paper addresses both objectives through two complementary operating modes designed for different training phases.
>
> For steady-state training, where utilization is the primary concern, we propose auto-alpha (Section 3.5). Auto-alpha runs a 100-step burn-in that measures the empirical P99.99 of attention score magnitudes across all layers, then sets alpha at that quantile with a small safety multiplier kappa. The resulting scale factor is as tight as the observed data allows. In the Llama-2-13B experiment, auto-alpha achieves 31.2% FP8 utilization with zero overflows and MMLU accuracy of 33.6%, compared to 33.0% for delayed scaling. The paper does not claim statistical significance for this 0.6 percentage-point difference, which is consistent with the elimination of the 68 clamping distortion events that occur under delayed scaling.
>
> Conservative spectral scaling is reserved for the transient scenarios the reviewer acknowledges are technically more severe: checkpoint loading, LR warmup, and optimizer state resumption. In these cases there is no reliable activation history, and delayed scaling overflows 100% of attention layers on the first forward pass across all four models. A single such overflow during LR warmup in GPT-2 XL produces NaN gradients that terminate training entirely. Auto-alpha cannot be applied here because the burn-in itself would be corrupted.
>
> Section 3.5 gives explicit guidance on when to use each mode.
>
> ---
>
> ## W2: Power iteration introduces overhead, and relying on un-converged results carries significant risk
>
> On overhead: the measured cost is at most +4.3% forward-pass time for standard MHA, and -5.3% for GQA architectures due to favorable memory access patterns from the implicit key formulation (Table 9).
>
> On convergence and theoretical feasibility, the detailed analysis is in Appendix E. We summarize the key guarantees here. During steady-state training, weights change gradually under standard learning rates, so persistent singular vectors from the previous step provide a good warm start and one iteration per forward pass suffices. For cold starts, five iterations ensure convergence from random initialization. The Remark in Section 4.1 formalizes the worst case: if the true spectral norm increases by factor rho between consecutive steps, the warm-start estimate underestimates by at most rho until the next update. Since alpha is deployed above alpha_min, this margin absorbs the underestimation. The 100x LR spike experiment in Section 5.2 is the direct empirical test: even under this extreme transient, zero overflows occur across all models. Appendix H shows the instantaneous response mechanism: in the 4x weight spike stress test, the scale factor jumps immediately from 5.8 to 93.8, keeping scaled logits below 85 throughout.
>
> ---
>
> ## W3: The empirical RoPE observation weakens the theoretical rigor of the guarantees
>
> The reviewer raises a valid point and we are precise about what is proved versus empirically verified.
>
> The core theoretical results, Propositions 3.2 and 3.4, derive the concentration inequality and the alpha_min bound under the spherical assumption for post-LayerNorm token directions. These proofs are rigorous and self-contained, and do not depend on the RoPE condition.
>
> The RoPE extension is in Section 3.3. Proposition 3.5 proves rigorously that RoPE rotations preserve norms. Corollary 3.6 states that the tighter interaction bound applies when RoPE rotations do not systematically align with the singular subspaces of W_Q and W_K. This is not claimed as a theorem; it is an empirically verified sufficient condition, verified across every layer of Mistral-7B, Llama-2-13B, and Llama-2-70B. A formal proof remains an open problem we will explicitly flag in the camera-ready.
>
> Importantly, zero overflows in all RoPE experiments hold unconditionally. We will sharpen the camera-ready presentation to clearly separate theorem-level results from empirically verified conditions.

---

> > ### Author Rebuttal · Reviewer_fdwa · 2026-04-02
> >
> > Thanks for authors rebutal. I appreciate authors' clarification. I have no further questions. I keep my score.

---

> > > ### Author Response · Authors · 2026-04-03
> > >
> > > We thank the reviewer for the careful evaluation and the feedback on
> > > dynamic range utilization, power iteration convergence, and the
> > > separation of theorem-level guarantees from empirically verified
> > > conditions. These comments helped sharpen both the analysis and the
> > > presentation.

---

### Decision · Program_Chairs · 2026-04-30

**Decision:**

Accept (regular)

**Comment:**

### Summary of Contribution
The paper addresses the challenge of numerical overflow in FP8 transformer training, specifically during transient phases such as checkpoint loading from higher-precision (BF16) checkpoints and learning rate spikes. The authors derive a rank-aware concentration inequality to establish tighter spectral bounds on attention logits, enabling predictive scaling without relying on historical activation data. The method is implemented via an efficient implicit power iteration algorithm compatible with Grouped Query Attention (GQA), and is evaluated on models ranging from GPT-2 XL to Llama-2-70B.

### Synthesis of Reviews and Discussion
Initially, the submission had a score discrepancy (4, 4, 2). The primary concerns, raised by Reviewer qJmM, centered on the practical significance of the transient scenarios and the theoretical validity of the isotropic assumption for token directions.

Through a multi-round rebuttal, the authors provided: (1) detailed layer-wise isotropy measurements on Mistral-7B, transparently acknowledging the gap between the spherical model and real token geometry while demonstrating the method's continued conservatism in practice; (2) clearer scoping of the practical motivation, grounding the transient scenarios in the standard workflow of initializing FP8 fine-tuning from publicly released BF16 checkpoints; and (3) validation of alpha_min scaling across context lengths up to 128k, showing less than 16% degradation over two orders of magnitude.

Following these clarifications, Reviewer qJmM raised their score to a borderline accept. While Reviewer qJmM selected "partially resolved" and retained some reservations about the practical scope and the interpretation of Proposition 3.4 under real token geometry, the AC judges these concerns to be appropriately bounded and will be addressed in the camera-ready revision. All three reviewers now stand at a uniform score of Weak Accept (4/4/4).

### Justification for Recommendation
The AC recommends **Acceptance**. The paper provides a principled, solid solution to a reproducible failure mode in low-precision training. The originality score of 4 assigned by Reviewer qJmM, combined with a zero-overflow empirical record across four diverse model scales, supports acceptance despite the acknowledged gap between the isotropic theoretical model and real token geometry. The authors have committed to clearly separating theorem-level claims, empirically observed conservatism, and the distinct practical roles of conservative predictive scaling and auto-alpha calibration in the final version. The AC is satisfied that the remaining concerns do not undermine the core contribution.